# SUMOylation of SAMHD1 at Lysine 595 is required for HIV-1 restriction in non-cycling cells

Charlotte Martinat[1,11], Arthur Cormier[1,11], Joëlle Tobaly-Tapiero[1], Noé Palmic[1], Nicoletta Casartelli[2,3], Bijan Mahboubi[4], Si'Ana A. Coggins[4], Julian Buchrieser [2,3,5], Mirjana Persaud[6], Felipe Diaz-Griffero [6], Lucile Espert[7], Guillaume Bossis [8], Pascale Lesage [1], Olivier Schwartz [2,3], Baek Kim [4], Florence Margottin-Goguet [9], Ali Saïb[1] & Alessia Zamborlini [1,10]✉

SAMHD1 is a cellular triphosphohydrolase (dNTPase) proposed to inhibit HIV-1 reverse transcription in non-cycling immune cells by limiting the supply of the dNTP substrates. Yet, phosphorylation of T592 downregulates SAMHD1 antiviral activity, but not its dNTPase function, implying that additional mechanisms contribute to viral restriction. Here, we show that SAMHD1 is SUMOylated on residue K595, a modification that relies on the presence of a proximal SUMO-interacting motif (SIM). Loss of K595 SUMOylation suppresses the restriction activity of SAMHD1, even in the context of the constitutively active phospho-ablative T592A mutant but has no impact on dNTP depletion. Conversely, the artificial fusion of SUMO2 to a non-SUMOylatable inactive SAMHD1 variant restores its antiviral function, a phenotype that is reversed by the phosphomimetic $T_{592}E$ mutation. Collectively, our observations clearly establish that lack of T592 phosphorylation cannot fully account for the restriction activity of SAMHD1. We find that SUMOylation of K595 is required to stimulate a dNTPase-independent antiviral activity in non-cycling immune cells, an effect that is antagonized by cyclin/CDK-dependent phosphorylation of T592 in cycling cells.

[1] INSERM U944, CNRS UMR 7212, Genomes & Cell Biology of Disease Unit, Institut de Recherche Saint-Louis, Université de Paris, Hôpital Saint-Louis, Paris, France. [2] Institut Pasteur, Virus and Immunity Unit, CNRS-UMR3569, Paris, France. [3] Vaccine Research Institute, Créteil, France. [4] Emory School of Medicine, Atlanta, USA. [5] James Martin Stem Cell Facility, Sir William Dunn School of Pathology, University of Oxford, Oxford, UK. [6] Albert Einstein College of Medicine, Microbiology and Immunology, Bronx, NY, USA. [7] IRIM, University of Montpellier, UMR 9004 CNRS, Montpellier, France. [8] IGMM, Univ Montpellier, CNRS, Montpellier, France. [9] Université de Paris, Institut Cochin, INSERM, CNRS, PARIS, France. [10] Institute for Integrative Biology of the Cell (I2BC), CEA, CNRS, Univ. Paris-Sud, Université Paris-Saclay, Gif-sur-Yvette, France. [11] These authors contributed equally: Charlotte Martinat, Arthur Cormier. ✉email: alessia.zamborlini@universite-paris-saclay.fr

Sterile alpha-motif (SAM) and histidine-aspartate (HD) domain-containing protein 1 (SAMHD1) is a cellular tri-phosphohydrolase (dNTPase) that inhibits the replication of the human immunodeficiency virus type 1 (HIV-1) in non-cycling immune cells such as macrophages, monocytes, dendritic cells, and resting T4 lymphocytes[1–5]. This antiviral function is largely attributed to the ability of SAMHD1 to hydrolyze dNTPs into the desoxynucleoside and triphosphate components[6–8] thereby reducing the cellular dNTP supply below a threshold required for efficient reverse transcription of the viral RNA genome[9,10]. In contrast to HIV-1, the related HIV-2 virus counteracts this restriction by expressing the Vpx accessory protein, which promotes the degradation of SAMHD1 through the ubiquitin-proteasome system[3–5,11,12]. SAMHD1 depletion is accompanied by both dNTP pools expansion and increased cell permissiveness to HIV-1 infection[8,13] indicating that the dNTPase and restriction functions are linked.

SAMHD1 is an ubiquitous protein[2,14,15]. Yet, its anti-HIV-1 activity is witnessed only in non-cycling cells, pointing to the involvement of post-translational regulatory mechanisms. It is now well established that residue T592 is phosphorylated by the cyclin/CDK complexes during the G1/S transition[16–18] and dephosphorylated by members of the phosphoprotein phosphatase (PPP) family upon mitotic exit[19,20]. This modification likely enables SAMHD1 to promote the progression of replication forks in dividing cells[21]. Phosphorylation at T592 is weak to undetectable in non-cycling cells refractory to HIV-1[19,20,22], suggesting that only dephosphorylated SAMHD1 might be restriction-competent. Consistent with this model, mutation of T592 into D or E to mimic phosphorylation renders SAMHD1 antivirally inactive[17,18,23,24]. However, the phosphomimetic variants retain WT dNTPase function[18,24,25]. In the same line, SAMHD1 prevents dNTP pools expansion throughout the cell cycle, regardless of its phosphorylation status[20]. Altogether these data question whether the establishment of a SAMHD1-mediated antiviral state might only rely on dNTP depletion and/or regulation by T592 phosphorylation. Reports that SAMHD1 degrades the incoming viral RNA genome through a ribonuclease activity remain controversial[23,26–30], calling for additional investigations to clarify the molecular mechanisms underlying its viral restriction function.

Interestingly, SAMHD1 was a hit in recent large-scale proteomic studies investigating the cellular substrates of SUMOylation[31,32], a dynamic post-translational modification (PTM) and an important regulator of many fundamental cellular processes including immune responses[33]. SUMOylation consists of the conjugation of a single small ubiquitin-like modifier (SUMO) moiety or a polymeric SUMO chain to a protein substrate through the sequential action of a dedicated set of E1-activating, E2-conjugating, and E3-ligating enzymes. SUMOylation is reversed by SUMO-specific proteases (e.g., SENP)[34]. Human cells express three ubiquitous SUMO paralogs. SUMO1 shares ~50% of sequence identity with SUMO2 and SUMO3, which are ~90% similar and thus referred to as SUMO2/3[35]. SUMOylation often targets the Lysine (K) residue lying within the consensus motif φKxα (φ: hydrophobic amino acid, x: any amino acid and α: an acidic residue) that represents the binding site for the unique SUMO E2 conjugating enzyme Ubc9[36,37]. A proximal SUMO-interacting motif (SIM), which typically consists of a short stretch of surface-exposed aliphatic residues[38], might sometimes contribute to the recruitment and the optimal orientation of the SUMO-charged Ubc9, allowing an efficient transfer of SUMO to the substrate. A SIM might also constitute a binding interface for SUMO-conjugated partners that mediate the downstream consequences of SUMOylation.

In this study, we show that SAMHD1 undergoes SIM-mediated SUMOylation of the evolutionarily conserved K595 residue, which is part of the CDK-targeted motif driving T592 phosphorylation ($^{592}$TPQK$^{595}$). Preventing K595 SUMOylation by inactivating either the SUMO-consensus motif or the SIM ($^{488}$LLDV$^{501}$) or by deleting the C-terminal region of SAMHD1 invariably suppressed its restriction, but not dNTPase activity. This was true even when T592 was dephosphorylated and therefore SAMHD1 expected to be antivirally active. These observations suggest that the status of T592 phosphorylation cannot fully account for the regulation of the restriction activity of SAMHD1. The artificial fusion of SUMO2 to an inactive C-terminal truncation mutant (lacking both T592 phosphorylation and K595 SUMOylation) restored viral restriction further supporting the requirement of SUMO conjugation to K595 for the establishment of a SAMHD1-dependent antiviral state in non-cycling immune cells.

## Results

### SAMHD1 is a SUMO target in both dividing and differentiated cells.

To investigate if SAMHD1 is a SUMO substrate, we used a 293T cell-based assay where we expressed HA-SAMHD1 together with each 6xHis-tagged SUMO paralog and the SUMO E2 conjugating enzyme Ubc9. Cells were treated or not with the proteasome inhibitor MG132, to favor the accumulation of SUMO-conjugated proteins[39]. Next, samples were lysed in denaturing conditions to inhibit the highly active SUMO proteases and preserve SUMOylation. Following enrichment of SUMOylated proteins by histidine affinity, the fraction of SUMO-modified SAMHD1 was detected with an anti-HA antibody. A ~100 kDa band, consistent with the expected size of SAMHD1 conjugated to a SUMO moiety was visualized in SUMO2- and SUMO3-expressing cells, at baseline (Fig. 1a, lanes 3 and 4). Proteasome inhibition caused an accumulation of high molecular-weight SAMHD1 species, which are likely SUMO chain conjugates, pointing to a potential role of SUMOylation in the control of the protein turnover (Fig. 1a, lanes 7 and 8). Modified SAMHD1 forms were undetectable in cells expressing SUMO1 or transfected with the empty plasmid, although the expression levels of SAMHD1 and SUMO isoforms were similar in all samples (Fig. 1a, lanes 1, 2, 5, and 6). Densitometric analyses showed that on average 12% of SAMHD1 is mono-SUMOylated (Fig. 1a and Supplementary Fig. 1). We also assessed the modification of SAMHD1 by endogenous SUMO paralogues. To this aim, 293T cells expressing HA-SAMHD1 were lyzed in stringent conditions, followed by immunoprecipitation on HA-matrix beads and immunoblotting with antibodies directed against SUMO1 or SUMO2/3. Consistent with the previous experiment, SUMO-conjugates were detected in the presence, but not in the absence of HA-SAMHD1 expression (Fig. 1b, lane 1 and 2; SUMOylation-deficient SIM2m and QQN variants are described below).

Since the antiviral effects of SAMHD1 are only witnessed in non-dividing immune cells, we next investigate its SUMOylation in differentiated myeloid cells. First, we stably expressed HA-SAMHD1 by lentiviral transduction in the human monocytic U937 cell line, which lacks measurable levels of endogenous SAMHD1 and acquires a macrophage-like phenotype when exposed to phorbol 12-myristate 13-acetate (PMA). Following denaturing lysis and enrichment on HA-matrix beads, we visualized high-molecular weight species reactive to anti-SUMO isoform-specific antibodies in SAMHD1-expressing, but not control cells (Fig. 1c). We also immunoprecipitated endogenous SAMHD1 from human monocytic THP1 cells differentiated into macrophage-like cells by PMA treatment. THP1 cells where

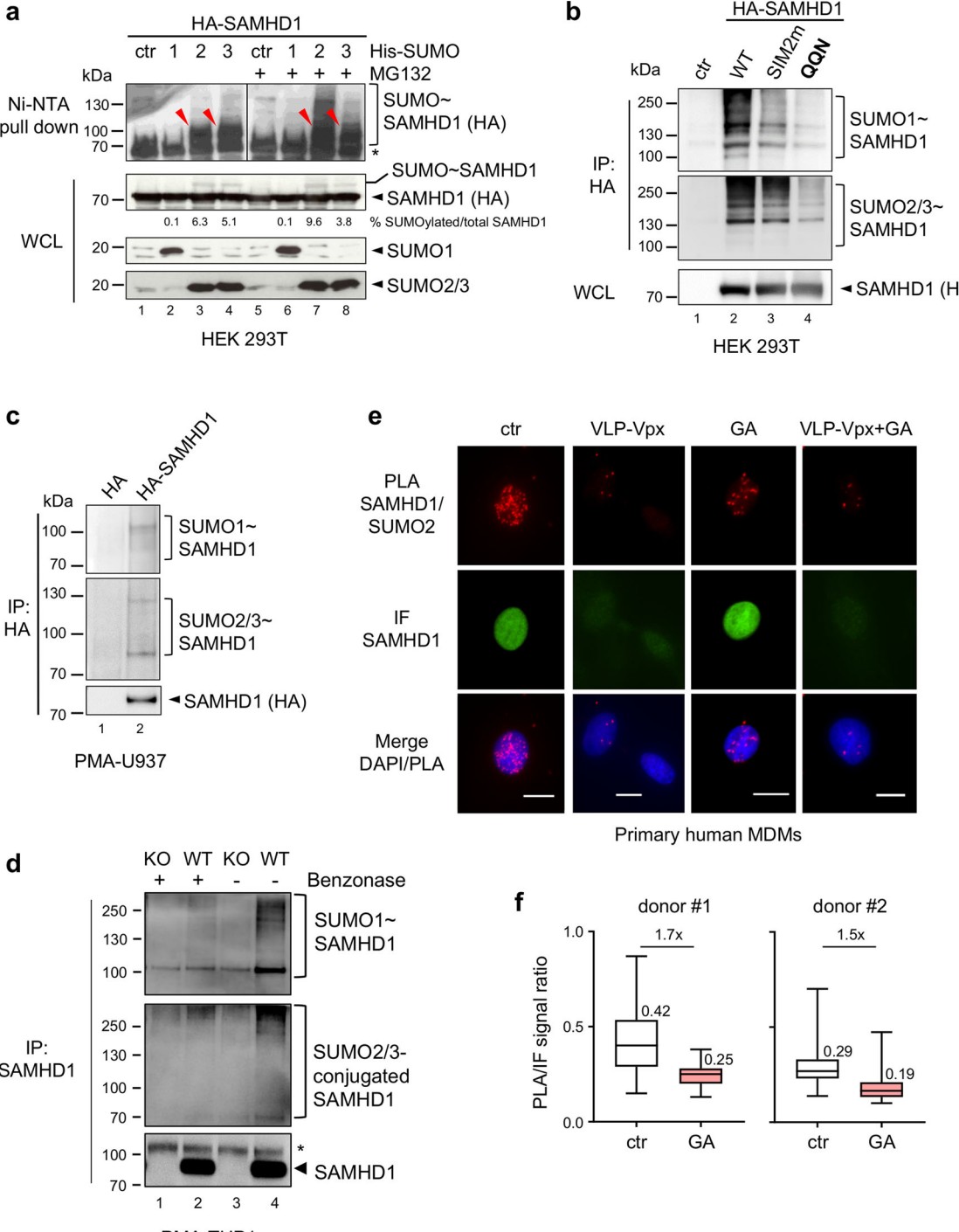

the *SAMHD1* gene was knocked out using the CRISPR/Cas9 technology served as negative control. Slow migrating bands were detected using anti-SUMO isoform-specific antibodies in WT but not SAMHD1-KO THP1 cells (Fig. 1d, lane 1 and 2). Notably, accumulation of the SUMOylated species was enhanced when lysis was carried out in benzonase-free buffer, indicating SAMHD1-SUMO association might be promoted by nucleic acids (Fig. 1d, lane 3 and 4).

To extend these findings, we used the proximity-ligation assay (PLA)[40,41] to analyze the interaction between SAMHD1 and SUMO in primary human monocyte-derived macrophages (MDMs). Fluorescent dots indicative of the SAMHD1-SUMO2/ 3 association were visualized mainly in the nucleus of untreated

cells (Fig. 1e, ctr panels), where both SAMHD1 and the SUMO machinery are enriched. The PLA signal intensity decreased by ~12-fold upon treatment with virus-like particles harboring Vpx (VLP-Vpx) to induce SAMHD1 degradation, which validates the specificity of the SAMHD1-SUMO2/3 proximity labeling (Fig. 1e, VLP-Vpx panels, and Supplementary Fig. 2a, middle panel). We also found that a 2-h incubation with ginkgolic acid (GA), which blocks SUMOylation by inhibiting the E1 SUMO-activating enzyme[42], lowered the PLA-to-IF signal ratio by ~1.6-fold as compared to the untreated control (Fig. 1e, GA panels, 1f and Supplementary Fig. 2a, left panel), while having no effect in VLP-Vpx pre-treated cells (Fig. 1e, VLP-Vpx+GA panels, and Supplementary Fig. 2a, right panel). Similar results were obtained

**Fig. 1 SAMHD1 is SUMOylated both in dividing and non-dividing cells. a** HEK 293T expressing HA-SAMHD1 and N-terminal His-tagged SUMO1 (1), SUMO2 (2), SUMO3 (3) or the control empty plasmid (ctr) were treated with MG132 or left untreated. After 48 h, cells were lyzed in denaturing conditions and SUMO-conjugates were enriched on Ni-NTA beads. Proteins contained in the eluates or the whole-cell lysates (WCL) were visualized using anti-HA or anti-SUMO paralog-specific antibodies. A representative result is shown (n = 3). Arrowheads point to SAMHD1~SUMO conjugates. *, nonspecific binding of unmodified SAMHD1 on Ni-NTA beads. The intensity of SUMOylated and unmodified SAMHD1 bands in the WCL was quantified by densitometry with ImageJ software. **b** HEK 293T cells expressing HA-SAMHD1 WT or SUMOylation-deficient SIM2m and QQN variants were lysed in denaturing buffer. After dilution, SAMHD1 and its post-translational derivatives were immunoprecipitated on HA-matrix beads and analyzed as in a. A representative result is shown (n = 2). **c** SUMOylation of HA-SAMHD1 stably expressed in differentiated U937 cells was assessed as in (**b**). **d** PMA-treated THP1 cells were lyzed in buffer supplemented with N-ethylmaleimide (NEM, 20 mM), Iodoacetamide (IAA, 5 mM) and benzonase (+, 50 U/mL) or not (−), followed by immunoprecipitation on beads coated with mouse anti-SAMHD1 antibodies. Immunoblotting was performed with anti-SAMHD1 or anti-SUMO isoform-specific antibodies. SAMHD1-KO THP1 cells (KO) were used as negative control. A representative result is shown (n = 2). *, nonspecific band. **e** Primary monocyte-derived macrophages (MDMs) generated from healthy donors (same as Fig. 6c) were incubated with virus-like particles containing Vpx (VLP-Vpx), or not, for 24 h and then treated with ginkgolic acid (GA, 100 µM) or DMSO (ctr) for 2 h. After fixation, cells were probed with rabbit anti-SAMHD1 and mouse anti-SUMO2/3 antibodies before being processed for Proximity Ligation Assay (PLA panels, red dots). SAMHD1 localization was analyzed using an anti-isotype secondary antibody coupled to the Alexa$_{488}$ dye (IF panels, green). Nuclei were stained with DAPI. Scale bar = 10 µm. Representative images of MDMs from n = 2 healthy donors are shown. **f** Box plots show the ratio between the PLA and IF signals that were measured for individual cells using the Icy software [donor 1: NT (96), AG (105); donor 2: NT (95), GA (113)]. Boxes extend from the 25th to 75th percentiles, the middle line is the median and whiskers go down to the smallest and up to the largest value. Mean values are indicated.

in differentiated THP1 cells exposed to GA or its structurally related analog anacardic acid (AA)[42] (Supplementary Figs. 2b, c). SAMHD1 localization (Fig. 1e and Supplementary Fig. 2b) and general expression (Supplementary Fig. 2a, d) as well as the global amount of SUMO2/3-conjugates (Supplementary Fig. 2d) were not detectably altered under these experimental conditions. An interaction between SAMHD1 and SUMO1 was also visualized in the nucleus of differentiated THP1, but not SAMHD1-negative U937 cells (Supplementary Fig. 3). Altogether, these data show that SAMHD1 is a SUMO target in both cycling and differentiated cells and suggest that this modification occurs in the nucleus.

**Lysine residues at position 469, 595, and 622 are the main SUMOylation sites of SAMHD1.** Among several potential SUMO-attachment sites identified by high-resolution proteomic studies in human SAMHD1, residues K469, K595 and K622 were both the most frequent and most abundantly modified hits (Supplementary Table 1). Protein sequence alignment shows that the position corresponding to amino acid 595 of human SAMHD1, which is the last residue of the CDK-targeted $^{592}$TPQK$^{595}$ motif (general consensus [S/T]-P-x-[K/R][43]), is invariably occupied by K, except for the murine isoform 2 (Supplementary Fig. 4). Conversely, K469 and K622 are conserved among primate orthologs, with the former also found in prosimian, equine, and koala isoforms (Supplementary Fig. 4). To confirm that the identified sites are modified by SUMO, we performed the 293T-based SUMOylation assay using SAMHD1 mutants where the candidate K residues were changed into either Arginine (R), to preserve a basic character, or Alanine (A) (Fig. 2a). Alternatively, we mutated the acidic residue at position +2 of the SUMO-acceptor K residue that is essential for the recruitment of Ubc9, the unique E2 SUMO-conjugating enzyme[36,37] (Fig. 2a). We focused our analyses on the SUMO2 paralog because (i) the pool of SUMO2 and SUMO3 available for conjugation exceeds that of SUMO1[44] and (ii) SUMO2 and SUMO3 differ only by three amino acids and are undistinguishable with existing antibodies[45]. Mutation of individual amino acids had a negligible effect on the electrophoretic mobility of SAMHD1 SUMO-conjugates both at baseline (Fig. 2b) and upon proteasome inhibition (Supplementary Fig. 5a) suggesting that multiple sites are simultaneously modified. To test this idea, we monitored the SUMOylation pattern of SAMHD1 variants where candidate sites were inactivated in various combination. Hereafter, these mutants are named by a three-letter code corresponding to the residues found at position 469, 595, and 622

(where K was replaced by either R or A) or 471, 597, and 624 (where E or D were replaced by Q or N, respectively). The simultaneous K$_{595}$A and K$_{622}$R changes (yielding the KAR variant, where K at position 469 is intact) prevented the formation of the ~100 kDa band seen with WT SAMHD1 and likely representing a mono-SUMOylated form (Fig. 2c, compare lanes 2 and 3). Thus, both K595 and K622 appear to be modified by a single SUMO moiety. Consistently, the mono-SUMOylated SAMHD1 form was detected when the SUMO site centered on either K595 or K622 was intact (corresponding to mutants RKR and QEN or RAK, respectively), although to a weaker extent relative to the WT protein (Fig. 2C, compare lanes 2 and 4, and lanes 7 and 8 to 6, densitometric quantification in Supplementary Fig. 1). We also analyzed the SUMOylation profile of SAMHD1 double mutants upon MG132 treatment. The KAR and RAK variants, harboring either intact K469 or K622, respectively, displayed an altered polySUMOylation pattern as compared to WT SAMHD1 (Supplementary Fig. 5b, compare lanes 3 and 4 to 2). Simultaneous substitution of R for K469 and K622 (yielding the RKR mutant) virtually abolished SAMHD1 polySUMOylation (Supplementary Fig. 5b, compare lanes 7 and 8 to 6). The concomitant E471Q and D624N changes (yielding the QEN mutant) had analogous consequences (Supplementary Fig. 5b, compare lanes 6 and 8). These results indicate that K469 and K622 are the major target sites for SUMO chains which accumulate upon MG132 treatment. Finally, we established that inactivation of the three SUMO-acceptor sites of SAMHD1 (RAR and QQN variants) strongly hampered the formation of slow migrating bands when SUMOs were expressed either ectopically (Fig. 2c and Supplementary Fig. 5b, lanes 9 and 10) or at endogenous levels (Fig. 1b, compare lanes 2 and 4). Overall, these results confirm that residues K469, K595, and K622 are the principal SUMOylation sites of SAMHD1.

**SAMHD1 mutants defective for K595 SUMOylation lose their HIV-1 restriction activity.** To assess the requirement of SUMO conjugation for viral restriction, we stably expressed WT or SUMOylation-site SAMHD1 mutants, in SAMHD1-negative monocytic U937 cells. Following cell differentiation by PMA treatment, all SAMHD1 mutants were enriched in the nucleus (Supplementary Fig. 6a) and displayed WT-like expression levels (Supplementary Figs. 6b, c). Next, we challenged the differentiated U937 cell lines with a VSVg-pseudotyped HIV-1 virus expressing *EGFP* as a reporter gene (VSVg/HIV-1∆Env*EGFP*) and quantified the fraction of infected cells 48 h later by flow cytometry (Fig. 3a). As previously reported, expression of WT

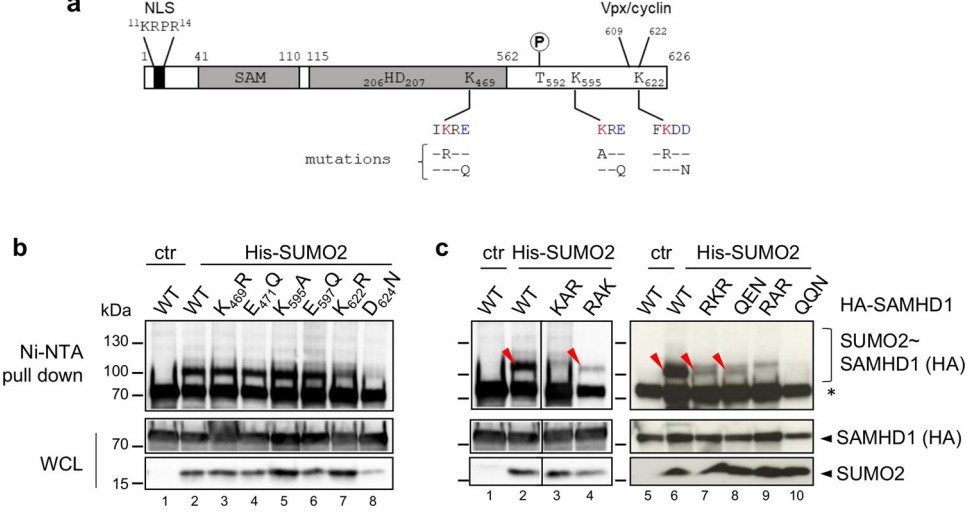

**Fig. 2 Residues K469, K595, and K622 are major SUMO conjugation sites of SAMHD1. a** Schematic representation of human SAMHD1 showing the nuclear localization signal (NLS), Sterile Alpha Motif (SAM) and Histidine/Aspartate (HD) domains, phosphorylatable T592 residue, and the C-terminal binding site for Vpx/cyclin A2. The position of the three putative SUMO consensus motifs is indicated, with the SUMO acceptor K and the acidic amino acids colored in red and blue, respectively. Residue substitutions are described below each SUMOylation site (mutations were done on single or multiple sites as described in the text). **b** HEK 293T cells overexpressing HA-SAMHD1 WT or single-site or **c** multiple SUMO-site mutants, Ubc9 and His-SUMO2 were processed as in Fig. 1a. Lanes 1 to 4 in panel C are derived from the same blot but were not adjacently loaded. WCL whole-cell lysate. *, nonspecific binding of unmodified SAMHD1 on Ni-NTA beads. The red arrowheads indicate the ~100 kDa band corresponding to mono-SUMOylated SAMHD1 species. Results of one representative experiment are shown ($n \geq 2$).

SAMHD1 rendered U937 cells resistant to HIV-1 infection, while the phosphomimetic $T_{592}E$ mutant failed to do so (Fig. 3b). Simultaneous substitution of R for the three major SUMO-acceptor K residues also abrogated SAMHD1-mediated restriction (RRR mutant, Fig. 3b, and Supplementary Fig. 6d). Similarly, preventing SUMOylation by mutation of the acidic amino acids within the corresponding SUMO consensus motifs rendered SAMHD1 restriction defective (QQN mutant, Fig. 3b). These observations strongly indicate that the regulation of SAMHD1 antiviral activity relies on SUMOylation, but not on other K-directed PTMs (i.e., ubiquitinylation, acetylation) that might target these sites. As SAMHD1 variants impaired for SUMO-conjugation to K469 and/or K622 efficiently blocked HIV-1 infection (RKR and QEN mutants, Fig. 3b and Supplementary Fig. 6d, e), we deduced that SUMOylation of K595 might be crucial for viral restriction by SAMHD1. Consistent with this hypothesis, SAMHD1 mutants where K595 was changed into either A or R lacked antiviral activity (Fig. 3c). Importantly, substituting E597 with Q to prevent K595 SUMOylation had similar functional consequences (Fig. 3c). Of note, mutating the neighboring residue Q594 into N did not modify the restriction activity of SAMHD1 (Fig. 3c).

To elucidate the possible mechanisms underlying the loss-of-restriction phenotype of SAMHD1 variants lacking K595 SUMOylation, we assessed their dNTPase activity by measuring cellular dNTP levels. The concentration of dATP and dGTP (representative of the four dNTPs) of differentiated U937 cells dropped ~20-fold upon expression of WT SAMDH1 (Fig. 3d and Supplementary Fig. 6f). As previously shown[8,18,24,25], the phosphomimetic $T_{592}E$ variant was as potent as WT SAMHD1 (Fig. 3d and Supplementary Fig. 6f). Similarly, all the tested SUMOylation-deficient SAMHD1 mutants reduced the cellular dNTP pools to a WT extent, indicating that their dNTPase activity is intact (Fig. 3d and Supplementary Fig. 6f). These results indirectly demonstrate that the impaired antiviral function of SAMHD1 variants lacking K595 SUMOylation is due to neither defective oligomerization nor improper folding. In conclusion, impairing SUMO conjugation to K595 compromises the antiviral

activity of SAMHD1 but not its dNTPase function, a phenotype that mirrors the effects of the phosphomimetic $T_{592}E$ mutation.

**Both K595 SUMOylation and viral restriction rely on the SIM2 motif.** In silico analysis of human SAMHD1 sequence with the bioinformatic predictor JASSA[46] highlighted the presence of three potential SIMs, suggesting that SAMHD1 could interact non-covalently with SUMO (Supplementary Fig. 7a). SIM1 ($_{62}PVLL_{65}$) lies within the N-terminal SAM domain while SIM2 ($_{488}LLDV_{491}$) and SIM3 (spanning the overlapping $_{499}IVDV_{501}$ and $_{500}VDVI_{502}$ sequences) are found in the C-terminal half of the protein (Fig. 4a). Protein sequence alignment revealed that SIM1 is present in SAMHD1 orthologs from Hominids, SIM2 also in Old-World Monkey isoforms, while SIM3 is highly conserved along evolution (Supplementary Fig. 4). By mapping the position of the putative C-terminal SIMs on the crystal structure of the HD domain of SAMHD1, we found that SIM3 is buried within the globular fold of the protein, while SIM2 is surface-exposed and near the SUMOylatable K595 residue (Fig. 4b), making it a more favorable candidate for functional studies. We first established that endogenous SAMHD1 was enriched on beads coupled to SUMO1 and, to a greater extent, SUMO2, but not uncoupled beads incubated with the lysate of differentiated THP1 cells (Fig. 4c). Next, we assessed the implication of SIM2 for the SAMHD1-SUMO binding. We found that mutating the LLDV sequence into AADA (yielding the SIM2m variant) resulted in a weaker association between SAMHD1 and SUMO2 in both pull-down (Fig. 4d) and PLA tests (Supplementary Fig. 7b, c).

The existence of a non-covalent interaction between SAMHD1 and SUMO proteins mediated by SIM2 prompted us to investigate the possible implications for viral restriction using the U937 cell-based assay described above. SAMHD1 SIM2m variant had defective anti-HIV-1 activity (Fig. 4e). Mutation of SIM2 also rendered SAMHD1 unable to restrict HIV-2ΔVpx (Supplementary Fig. 8a) without affecting its localization (Supplementary Fig. 7b), expression levels (Supplementary Fig. 8b, c) and capacity to reduce

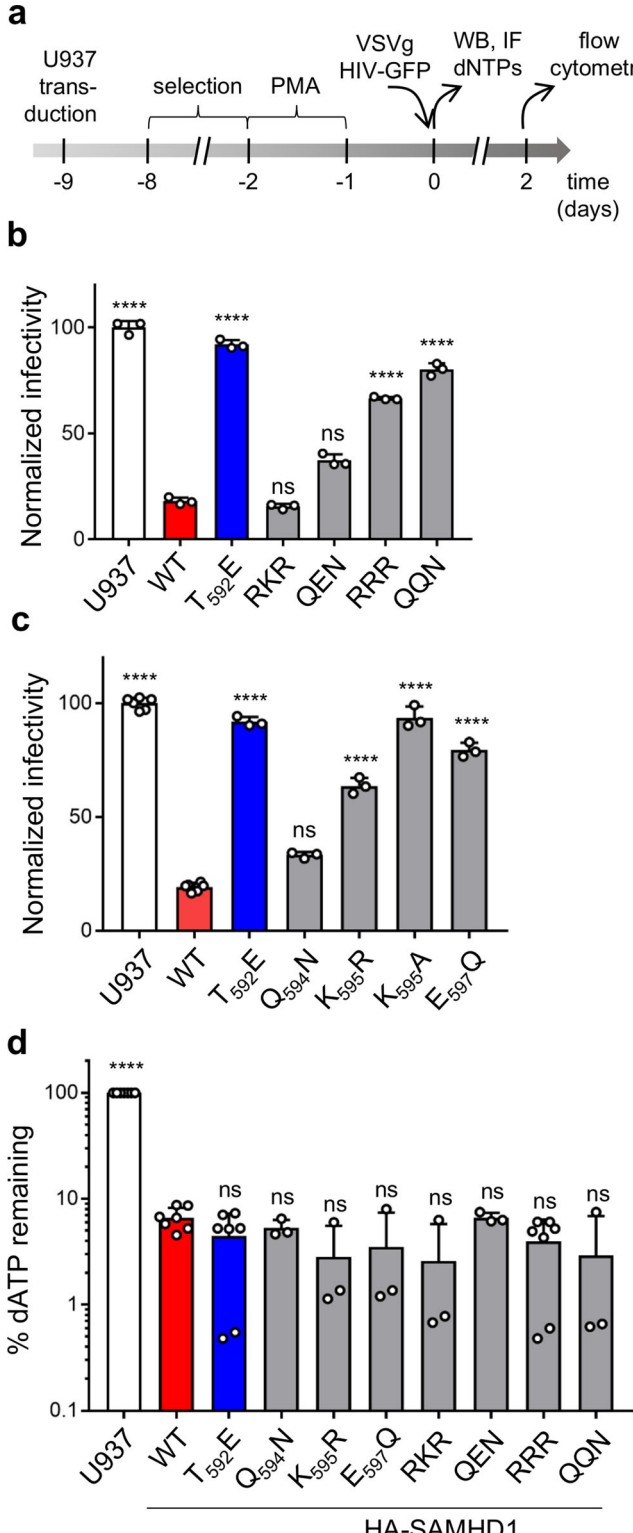

**Fig. 3 SAMHD1 mutants impaired for SUMOylation on K595 are antivirally inactive but efficiently deplete the cellular dNTP pools. a** U937 cell lines stably expressing HA-SAMHD1 WT or mutants were treated according to the experimental outline. **b** Differentiated U937 cell lines expressing the indicated multiple (≥3 independently generated cell lines) or **c** single SUMO-site SAMHD1 variants (4 independently generated cell lines) were infected with the VSVg/HIV-1ΔEnv*EGFP* virus and analyzed by flow cytometry 48 h later. The infection rate of parental U937 cells was set to 100. Data (mean ± SD) from one representative experiment performed in three technical replicates are shown (for (**b**): $n = 10$ for U937 and WT, $n = 9$ for T$_{592}$E, 4 for RKR and QEN, $n = 7$ for RRR, 3 for QQN; for (**c**): $n = 9$ for U937, WT, T$_{592}$E, K$_{595}$R, $n = 6$ for Q$_{594}$N and E$_{597}$Q, $n = 5$ for K$_{595}$A). Statistical significance was assessed by one-way ANOVA test with a Dunnett's multiple comparison post test. ****$p < 0.0001$. ns not significant, $p > 0.05$. d Cellular dATP levels were quantified in differentiated U937 cell lines by single nucleotide incorporation assay[9]. The dNTP levels (%) of SAMHD1-expressing cells were calculated relative to those of parental U937 set to 100. Bars show the mean ± SD ($n = 5$ for U937, WT, T$_{592}$E, RRR; $n = 3$ for the other mutants). Statistical significance was assessed by one-way ANOVA test with a Dunnett's multiple comparison post test. ****$p < 0.0001$. ns not significant, $p > 0.05$.

As mutation of SIM2 and/or the adjacent SUMOylation motif harboring K595 abrogated SAMHD1 restriction activity, we postulated the existence of a functional connection between the two sites. Indeed, SIM2 might provide an extended binding interface that stabilizes the association between SAMHD1 and the SUMO-charged Ubc9 promoting the efficient transfer of SUMO to K595, which lies within a minimal SUMOylation site (KxE)[34]. Consistent with our hypothesis, we confirmed that the LLDV/AADA substitution in the context of the SAMHD1 RKR variant virtually abolished the formation of the ~100 kDa band corresponding to K595 SUMOylation by performing both immunoprecipitation and histidine affinity purification assays (Fig. 4f, compare lanes 2 and 3). Conversely, inactivation of SIM2 barely modified the global SUMOylation profile of WT SAMHD1 in conditions of either ectopic (Supplementary Fig. 8e, compare lanes 2 and 3) or endogenous expression of SUMO isoforms (Fig. 1b, compare lanes 2 and 3). In conclusion, the integrity of the surface-exposed SIM2 is essential for both SUMO conjugation to K595 and viral restriction, providing converging evidence that human SAMHD1 requires K595 SUMOylation to be restriction competent.

**T592 phosphorylation and K595 SUMOylation are independent events.** SAMHD1 mutants lacking SUMO conjugation to K595 mirrored the loss-of-restriction phenotype of the phosphomimetic T$_{592}$E variant, raising the possibility that SUMOylation and phosphorylation of these adjacent sites might influence each other. To address this point, we expressed HA-SAMHD1 WT and/or His-SUMO2 in HEK 293T cells and performed histidine affinity purification or immunoprecipitation tests. A ~100 kDa band, corresponding to mono-SUMOylated SAMHD1, was readily detected with an antibody directed against its T592 phosphorylated species in the presence, but not in the absence, of His-SUMO2 expression (Supplementary Fig. 9a, b). Densitometric analysis revealed that ~6% of T592 phosphorylated SAMHD1 is simultaneously SUMOylated (Supplementary Fig. 8b). A ~100 kDa band reactive to the anti-phospho T592 species-specific antibody was also visualized in cells co-expressing HA-SAMHD1 RKR and His-SUMO2, but not the corresponding empty vector (Supplementary Fig. 9c), consistent with the identification of the corresponding SUMO-phosphopeptide in a previous proteomic screen[47]

cellular dNTP concentrations (Supplementary Fig. 8d). SIM2 is located near residue T592, which phosphorylation is recognized as a major regulator of SAMHD1 antiviral function[17,18]. Therefore, we used an anti-phospho-T592 species-specific antibody to monitor the degree of modification of SAMHD1 SIM2m mutant expressed either stably in differentiated U937 cells or transiently in cycling 293T cells. In both cell types, SAMHD1 SIM2m variant was phosphorylated at levels comparable to that of WT SAMHD1 (Supplementary Fig. 8b, c).

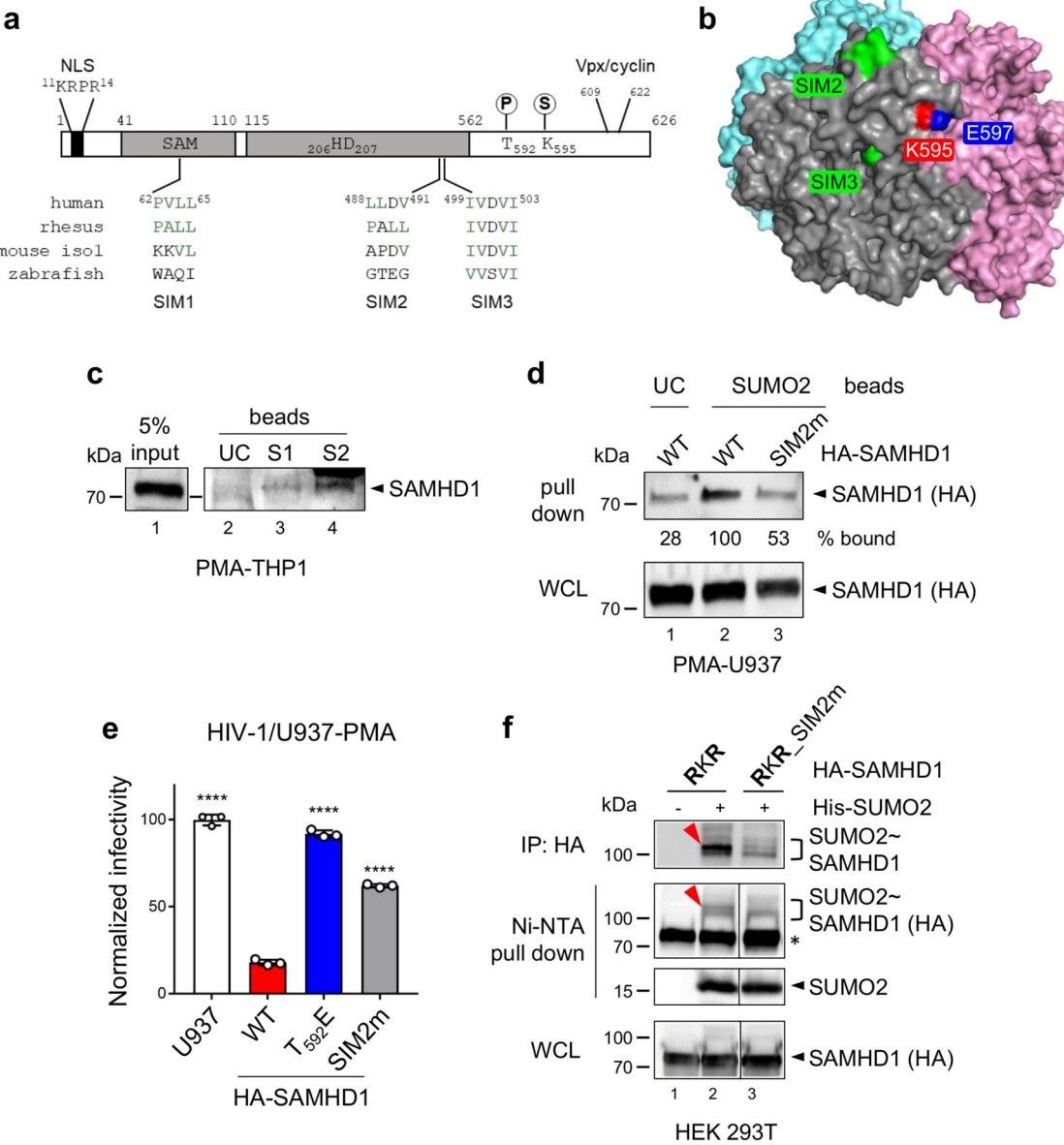

**Fig. 4 Integrity of SAMHD1 SIM2 is required for both HIV-1 restriction and K595 SUMOylation. a** Schematic representation of human SAMHD1 (NP_056289.2), showing the position and sequence of the putative SIMs and alignments with corresponding sequences of isoforms from Rhesus macaque (NP_001258571.1), mouse (NP_061339.3, isoform 1), and zebrafish (NP_001153405.1). **b** Position of SIM2 and SIM3 (green), K595 (red), and E597 (blue) within one protomer of human SAMHD1 tetramer (PDB: 4BZC). **c** The lysate of differentiated THP1 cells was split in equal aliquots that were incubated with agarose beads coated with either human recombinant SUMO1 (S1) or SUMO2 (S2), or uncoated beads (UC) as control. Proteins from the input and the eluates were separated by migration on a 4–15% SDS-PAGE gel and, next, visualized by immunoblotting using antibodies against SAMHD1. Results of one representative experiment are shown ($n = 3$). **d** The lysate of differentiated U937 cells expressing WT or SIM2m SAMHD1 variants was incubated with SUMO2-coated agarose beads and treated as in (**c**). The band intensities were quantified with ImageJ software. Results of one representative experiment are shown ($n = 2$). **e** U937 cells stably expressing the indicated HA-SAMHD1 variants (three independently generated cell lines) were infected with the VSVg/HIV-1ΔEnv*EGFP* virus and analyzed as in Fig. 3b. The infection rate of parental U937 cells was set to 100. Data (mean ± SD) from one representative experiment performed in three technical replicates are shown ($n = 5$). **f** HEK 293T cells overexpressing HA-SAMHD1 RKR mutant, Ubc9, and His-SUMO2 were lyzed in denaturing conditions and split in two equal aliquots that were subject to Ni-NTA pull down or affinity purification on HA-matrix beads. Eluted proteins were analyzed as described in Fig. 1a. WCL whole-cell lysate. Results of one representative experiment are shown ($n = 2$). The red arrowhead highlights the SUMO-conjugated K595 SAMHD1 species. *, nonspecific binding of unmodified SAMHD1 on Ni-NTA beads. Lanes 1–3 are derived from the same blot but were not adjacently loaded.

(Supplementary Table S1). To extend these findings, we studied the SUMOylation profile of SAMHD1 RKR mutants where T592 was simultaneously changed into A or E, to mimic the absence or presence of a phosphate group, respectively, and found it undistinguishable from that of the WT counterpart in both Ni-NTA pull down and immunoprecipitation assays

(Supplementary Fig. 10a, b). Thus, the phosphorylation status of T592 does not seem to influence K595 SUMOylation.

We then analyzed the degree of phosphorylation of WT and SUMOylation-deficient SAMHD1 variants expressed in 293T cells using the phospho-T592 species-specific antibody. The $E_{597}Q$ change did not detectably modify the ratio of the phosphorylated

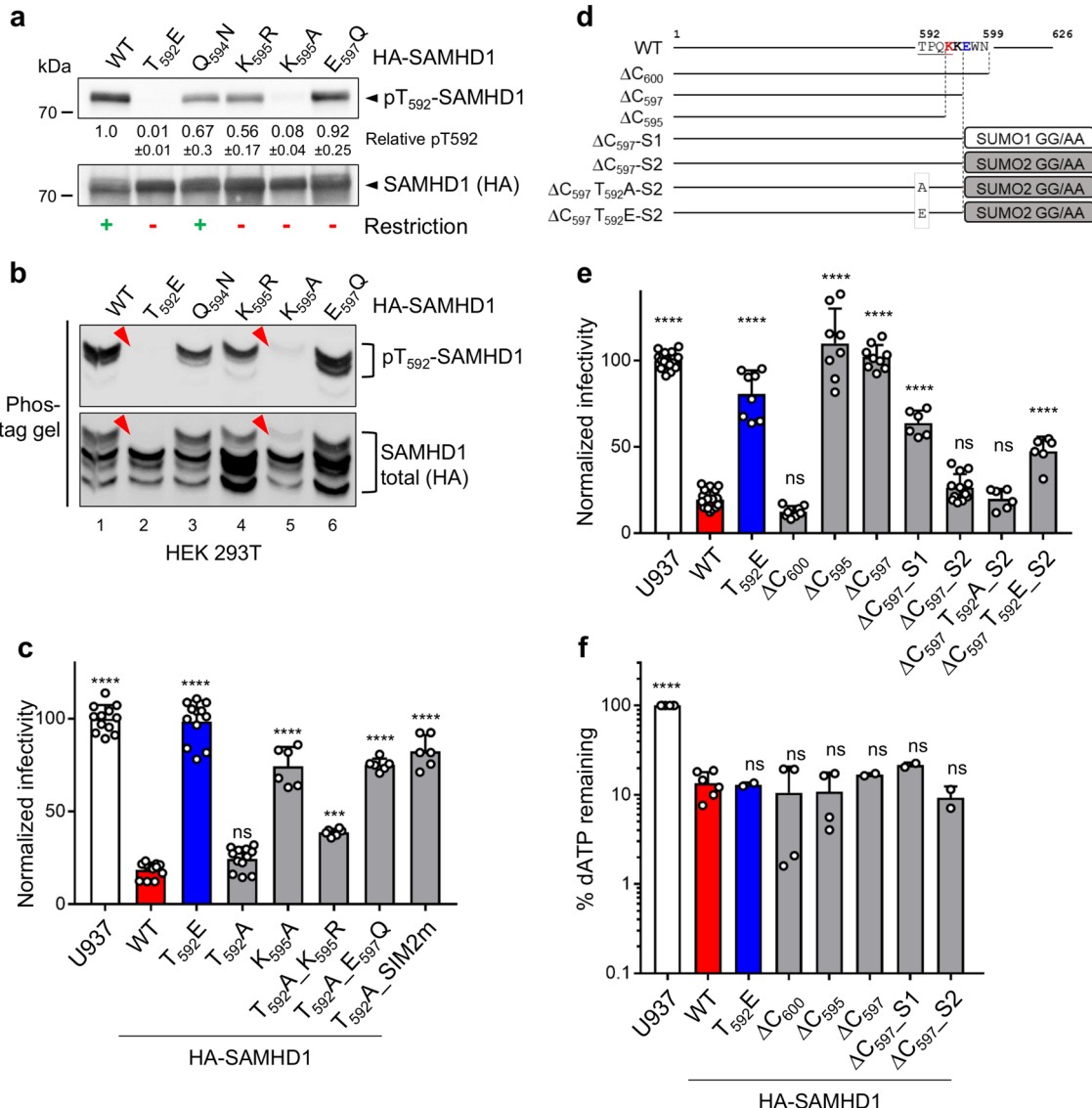

**Fig. 5 Dephosphorylated T592 is not sufficient to render SAMHD1 antivirally active, concomitant SUMOylation of K595 is required. a** Proteins (10 µg total proteins/line) contained in the crude extract of 293 T cells overexpressing HA-SAMHD1 variants were loaded on a 4–15% pre-casted SDS-PAGE gel. Immunoblotting was performed sequentially with an anti-pT592 or anti-HA antibody to detect phosphorylated or total SAMHD1 species, respectively. The band intensities were quantified by densitometry with ImageJ software and the pT592/total SAMHD1 ratio for WT SAMHD1-expressing cells was set to 1. One representative immunoblot is shown, while the quantification data represent the mean ± SD of all the experiments ($n \geq 2$). The ability of SAMHD1 mutants to restrict (green+) or not (red−) viral infection is indicated. **b** The same samples as in A were separated on a 7% Phos-tag™ SDS-PAGE gel. Arrowheads indicate that T592 phosphorylated SAMHD1 species which become undetectable upon $T_{592}E$ mutation. Results of one representative experiment are shown ($n = 3$). **c** U937 cell lines (2–4 independently generated cell lines) stably expressing the indicated HA-SAMHD1 mutants were differentiated and next infected with the VSVg/HIV-1ΔEnv*EGFP* virus and analyzed by flow cytometry 24 h later. The infection rate of parental U937 cells was set to 100. **b** Data (mean ± SD) from one representative experiment performed in three technical replicates are shown ($n = 4$ for U937, WT, $T_{592}E$, and $T_{592}A$; $n = 2$ for $K_{595}A$, $T_{592}A\_K_{595}R$, $T_{592}A\_E_{597}Q$, and $T_{592}A\_SIM2m$). Statistical significance was assessed by one-way ANOVA test with Dunnett's multiple comparison post test. ***$p < 0.001$, ****$p < 0.0001$. ns: not significant, $p > 0.05$. **d** A schematic representation of SAMHD1 variants harboring C-terminal deletions or expressed as SUMO1- or SUMO2-fusion proteins. The C-terminal diglycine motif of SUMO isoforms is changed into di-alanine to prevent conjugation (GG/AA). Residues of the CDK- and SUMO-consensus motif are underlined or bold and K595 and E597 are colored in red and blue, respectively. **e** U937 cell lines ($\geq 2$ independently generated cell lines) stably expressing the indicated HA-SAMHD1 mutants were analyzed as in (**c**). Statistical significance was assessed by one-way ANOVA test with Dunnett's multi-comparison post test. ****$p < 0.0001$. ns not significant, $p > 0.05$. **f** The levels of dATP were quantified and normalized as in Fig. 3d. The dNTP levels of U937 were set to 100%. Bars show the mean ± SD ($n \geq 2$). Statistical significance was assessed by one-way ANOVA test with Dunnett's multiple comparison post test. ****$p < 0.0001$. ns not significant, $p > 0.05$.

relative to the total SAMHD1 levels, indicating that SUMO-conjugation to K595 is not absolutely required for T592 phosphorylation. We also observed that the $Q_{594}N$ and $K_{595}R$ substitution reduced T592 phosphorylation by ~33% to 47%, respectively (Fig. 5a). Strikingly, the $K_{595}A$ mutation, which

rendered SAMHD1 antivirally inactive (Fig. 3c), caused a ~94% drop in T592 phosphorylation (Fig. 5a), likely by disrupting the $^{592}$TPQK$^{595}$ CDK-consensus sequence[43]. We obtained similar results in differentiated U937 cells (Supplementary Fig. 10c), in agreement with a previous report[22]. Lack of T592 phosphorylation

upon replacing K595 with A was confirmed by comparing the electrophoretic mobility of WT and SUMOylation-deficient SAMHD1 variants in a Phos-tag gel, ruling out the possibility that amino acids changes near T592 had altered the binding affinity of the phospho-specific antibody (Fig. 5b). Although a direct influence of SUMOylation on T592 phosphorylation cannot be fully excluded, it was not possible to define a hierarchical relationship between SUMO conjugation on K595 and T592 phosphorylation, implying that they are to some degree independent events.

**Modification of K595 by SUMO2 renders dephosphorylated SAMHD1 antivirally active.** Since the $K_{595}A$ mutant was hypo-phosphorylated but still lacked antiviral activity, we deduced that absence of T592 phosphorylation alone is insufficient to render SAMHD1 restriction competent and that SUMO conjugation to K595 is required. To substantiate this conclusion, we generated U937 cell lines stably expressing SAMHD1 variants bearing the phospho-ablative $T_{592}A$ change alone or together with mutations disrupting SUMO conjugation to K595 (Supplementary Fig. 11a) and tested their restriction activity. As previously reported[17,18,23,48], the $T_{592}A$ mutant hampered HIV-1 infection as efficiently as WT SAMHD1 (Fig. 5c). The concomitant $E_{597}Q$ mutation or inactivation of SIM2 rendered SAMHD1 $T_{592}A$ variant antivirally inactive to the same level as the individual $T_{592}E$ or $K_{595}A$ change (Fig. 5c). Intriguingly, the $T_{592}A\_K_{595}R$ SAMHD1 mutant displayed a milder, yet statistically significant, loss-of-restriction phenotype (Fig. 5c). We speculated that SAMHD1 harboring the conservative $K_{595}R$ substitution might still be able to recruit the SUMO-charged Ubc9, which might transfer SUMO to nearby K residues. In agreement, the $K_{596}R$ change slightly accentuated the restriction defect of the $T_{592}A\_K_{595}R$ variant (Supplementary Fig. 11b). Notably, five additional potentially SUMOylatable K, beside K596, lie in the region surrounding SIM2 and the SUMO-consensus motif centered on K595 (Supplementary Table 1 and Supplementary Fig. 11c).

To bring additional evidence for the implication of K595 SUMOylation in the SAMHD1-mediated antiviral mechanism, we undertook two complementary strategies. First, we assessed the consequences of progressively truncating the C-terminal region of SAMHD1 on the restriction activity. Deletion of the 26 C-terminal amino acids (yielding HA-SAMHD1 $\Delta C_{600}$) altered neither the integrity of the SUMO-consensus motif centered on K595 (Fig. 5d), nor antiviral activity (Fig. 5e). Conversely, SAMHD1 truncation mutants harboring partial (HA-SAMHD1 $\Delta C_{597}$) or complete deletion (HA-SAMHD1 $\Delta C_{595}$) of the SUMOylation site (Fig. 5d) was inactive against HIV-1 (Fig. 5e), as previously shown[23,49]. Next, we engineered two fusion proteins where the sequence of unconjugatable SUMO1 or SUMO2 was inserted in-frame at the C-terminus of the SAMHD1 $\Delta C_{597}$ variant to mimic a constitutively SUMOylated form (Fig. 5d). We reasoned that, if the defect was due to lack of K595 SUMOylation, the SUMO-fusion should rescue the antiviral function of the SAMHD1 C-terminal truncation mutant, as reported for CtIP-dependent DNA resection activity[50]. In agreement with this hypothesis, fusion of SUMO2, but not SUMO1, turned the restriction-defective SAMHD1 $\Delta C_{597}$ variant into a restriction-competent protein (Fig. 5e). Finally, we examined the impact of mutating T592 into A or E on the anti-HIV-1 function of the SAMHD1-SUMO2 chimera. As shown in Fig. 5e, the $T_{592}A$ change did not modify the restriction activity of the SAMHD1-SUMO2 fusion protein, while the phosphomimetic $T_{592}E$ mutation alleviated it, yet not as dramatically as in the context of WT SAMHD1. Parallel experiments confirmed that all the SAMHD1 variants reduced the cellular dNTP concentrations like the WT protein (Fig. 5f and Supplementary Fig. 12a) and displayed similar expression levels (Supplementary Fig. 12b). The

fusion of SUMO proteins promoted the accumulation of SAMHD1 in nuclear foci (Supplementary Fig. 12c). Intriguingly, it also restored phosphorylation of T592 in the context of a SAMHD1 C-terminal truncation mutant lacking the CDK consensus site (Supplementary Fig. 12b). Further investigations are warranted to assess whether SUMOylation of K595 plays a direct role in T592 phosphorylation, as shown for pRb[51], or whether increased phosphorylation is an indirect consequence of the artificial fusion of SUMO.

Collectively, our data demonstrate that, in non-cycling cells where the bulk of SAMHD1 harbors dephosphorylated T592, SUMOylation of K595 defines the fraction of the protein that is restriction competent. They also indicate that cyclin/CDK-dependent phosphorylation of T592 antagonizes the effect of K595 SUMOylation, relieving the SAMHD1-mediated restriction.

**SUMOylation inhibition increases the infectivity of SAMHD1-sensitive viruses in macrophages.** Having shown that K595 SUMOylation regulates the antiviral activity of SAMHD1 stably expressed in U937 cells, we undertook a drug-based approach to validate this finding on endogenous SAMHD1-expressing cells. We therefore exposed undifferentiated or PMA-treated THP1 cells to GA or AA, to suppress the activity of the E1 SUMO-activating enzyme[42] and hinder the SAMHD1-SUMO association (Supplementary Fig. 2b), before challenge with the VSVg/HIV-1ΔEnv*EGFP* virus. Cycling THP1 cells were readily permissive to HIV-1 and inhibition of SUMOylation did not alter this state (Fig. 6a), indirectly demonstrating that cell viability was not affected. The percentage of HIV-1-positive cells sharply decreased (~15-fold) following differentiation and was rescued to a moderate (~1.7 to 2-fold) but statistically significant extent by blocking the SUMO pathway (Fig. 6a). This effect was neither due to altered SAMHD1 localization (Supplementary Fig. 2b) and expression (Supplementary Fig. 2d), nor changes in the dNTP concentrations (Supplementary Fig. 13a). The implication of the SUMOylation process in the antiviral mechanism of SAMHD1 was further substantiated by finding that treatment with GA increased ~2.5-fold the permissiveness of differentiated THP1 cells to HIV-2ΔVpx without affecting the infectivity of the corresponding WT virus (Fig. 6b).

Finally, we investigated the outcome of GA treatment on the spread of the replication-competent HIV-1 AD8 macrophage-tropic virus in primary monocyte-derived macrophages (MDMs). Inhibition of SUMOylation increased the magnitude of HIV-1 viral particle release in cultures of MDMs generated from 3 different donors, although to a various degrees (Fig. 6c). In agreement with the results above, GA had no effects on viral replication when SAMHD1 had been previously degraded by VLP-Vpx (Fig. 6c). Inhibiting SUMOylation similarly favored HIV-1 spread in induced-pluripotent stem cells (iPSc)-derived macrophages (Supplementary Fig. 13b). Altogether, these observations confirm that the maintenance of a SAMHD1-dependent antiviral state in macrophages relies on a functional SUMO system.

## Discussion

It is widely accepted that the antiviral activity of SAMHD1 is downregulated by phosphorylation of residue T592 in actively dividing cells. Yet, T592 phosphorylation does not influence the dNTPase function, which is a central element of the restriction mechanism mediated by SAMHD1, implying that additional activities and/or regulations are at play. In this study, we show that SAMHD1 is SUMOylated and provide several lines of evidence that this modification is required to stimulate its antiviral activity in non-cycling immune cells. We established that

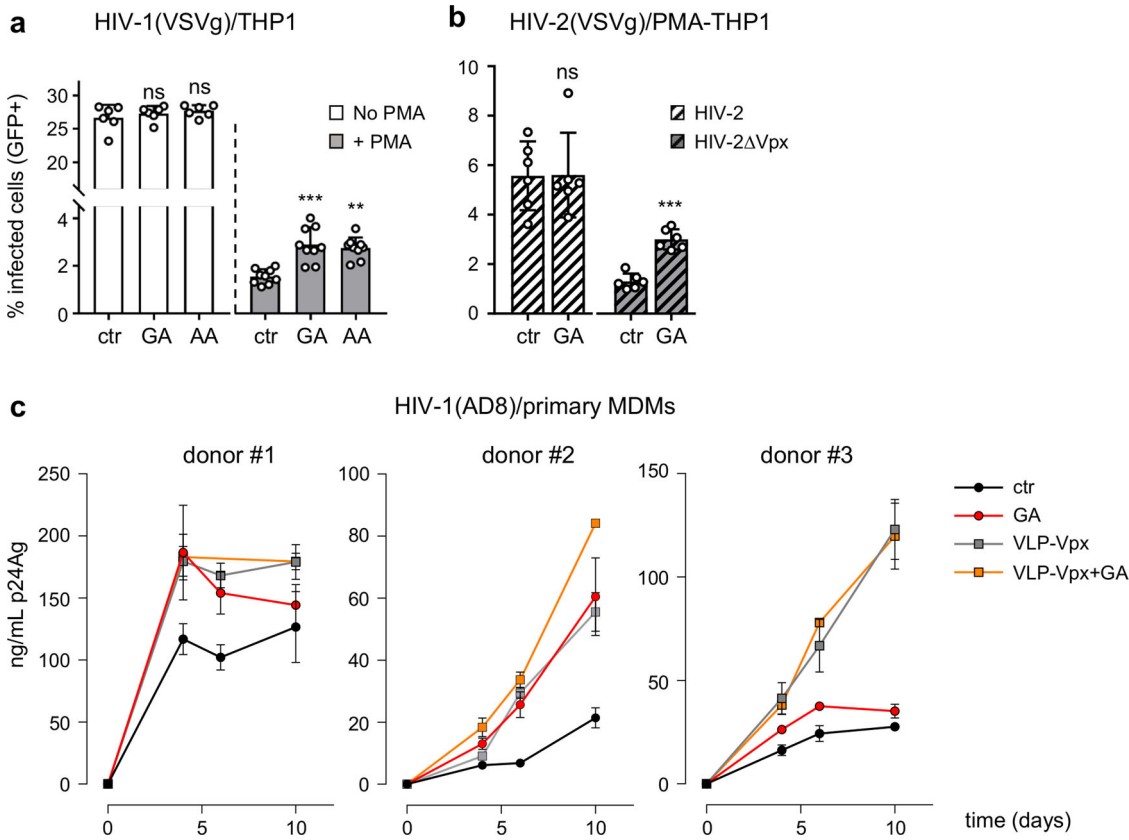

**Fig. 6 Infectivity of SAMHD1-sensitive viruses is enhanced by inhibition of SUMOylation in macrophages. a** THP1 cells were differentiated into macrophage-like cells by incubation with PMA (100 ng/mL, 24 h), or left untreated. Twenty-four hours later SUMOylation was inhibited by exposure to gingkolic acid (GA, 50 μM) or anacardic acid (AA, 50 μM) for 2 h before challenge with VSVg-pseudotyped HIV-1 (moi = 0.2), **b** HIV-2 or HIV-2ΔVpx (moi = 0.3) viruses harboring the *EGFP* reporter gene. The percentage of infected (GFP-positive) cells was measured by flow cytometry after 48 h. Data (mean ± SD) from one representative experiment performed in three technical replicates are shown (n ≥ 2). Statistical significance was assessed by two-way-ANOVA test with a Dunnett's multiple comparison post-test. **p < 0.01; ***p < 0.001; ns not significant, p > 0.05. **c** Human MDMs generated from healthy donors (same as Fig. 1d) were incubated with VLP harboring Vpx, or not, for 24 h and then exposed to GA or vehicle (DMSO, 2 h) before challenge with the AD8 HIV-1 strain (50 ng/mL p24Ag). Viral replication was monitored overtime by measuring the p24 antigen (p24Ag) released in the cell culture supernatant.

SAMHD1 harbors three major SUMO-attachment sites that map to residues K469, K595, and K622. Still, we cannot exclude that additional sites might be SUMOylated at a low level or under specific circumstances (Supplementary Table 1). By analyzing a large panel of SAMHD1 variants, where these amino acids were mutated individually or in various combinations, we determined that (i) SUMOylation occurs on multiple sites simultaneously, (ii) K595 and K622 undergo mono-SUMOylation (Fig. 2c and 7a) and (iii) K469 and K622 are targeted by SUMO chains, which accumulate upon inhibition of the proteasome (Supplementary Fig. 5b and Fig. 7a). Ubiquitination of K622 mediated by TRIM21, which belongs to a protein family comprising ligases with dual SUMO and Ubiquitin E3 activity[52], was recently proposed to favor proteasomal degradation of SAMHD1 in enterovirus-infected cells[53]. Whether Ubiquitination and SUMOylation of the same site cooperate to regulate the fate of SAMHD1 remains open for future studies. Combining biochemical and/or imaging approaches, we established that SAMHD1 is modified by SUMO in cycling 293T cells as well as differentiated cells of the myeloid lineage where its antiviral function is witnessed, including primary MDMs (Fig. 1 and Supplementary Fig. 2). These observations indicate that global SUMOylation of SAMHD1 is not influenced by the cell division status although site-specific variations depending on the

cellular context cannot be excluded. To add an extra level of complexity, we discovered that SAMHD1 harbors a surface-exposed SIM (named SIM2, aa 488–491), which promotes SUMO conjugation to the adjacent K595 residue likely by contributing to both the efficient recruitment and the optimal orientation of the SUMO-charged Ubc9[34]. Mediating a preferential non-covalent interaction between SAMHD1 and SUMO2, SIM2 might also dictate the selective modification of K595 by this paralog.

By interrogating the ability of SUMOylation-deficient SAMHD1 variants to inhibit HIV-1, we found that simultaneous mutation of the three SUMO-acceptor K residues abolishes viral restriction. The same was observed upon substitution of the acidic amino acids within the corresponding SUMO-consensus motifs, to prevent the recruitment of the SUMOylation machinery[36,37], strongly supporting the implication of SUMOylation, but not other K-directed PTMs, for the antiviral function of SAMHD1. Inhibiting SUMO conjugation to K595 in several ways (deleting the C-terminal region encompassing aa 595-626, mutating key residues of either the SUMO consensus motif or the SIM2) invariably recapitulated the loss-of-restriction phenotype. This was true even in the context of a constitutively active T₅₉₂A variant, indicating that dephosphorylated T592 alone cannot overcome the defect imposed by the absence of K595

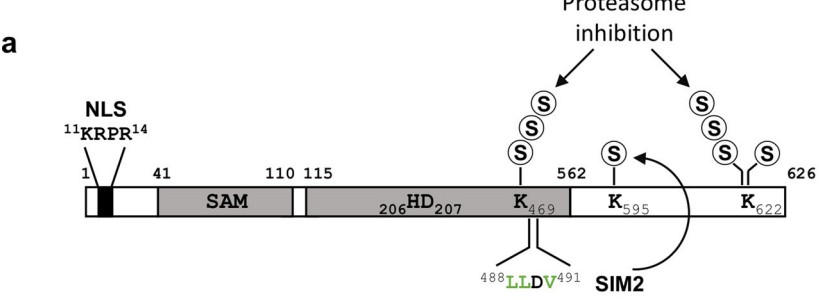

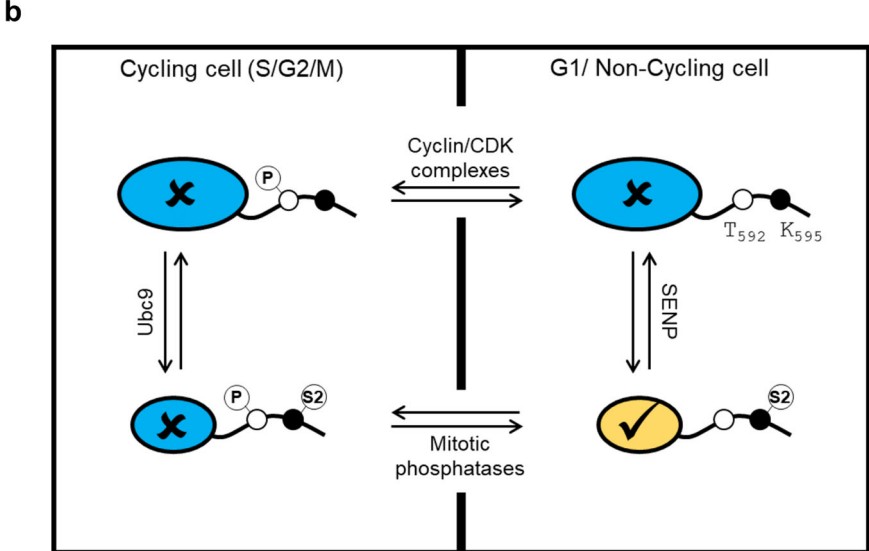

**Fig. 7 Model for the regulation of SAMHD1 antiviral activity by SUMOylation in non-dividing cells. a** Human SAMHD1 harbors three major SUMO-attachment sites: K595 and K622 undergo mono-SUMOylation, while K469 and K622 are targeted by SUMO chains, which accumulate upon inhibition of the proteasome. Human SAMHD1 also harbors the surface-exposed SIM2 sequence, which drives the modification of K595 by SUMO, likely with a preference for the SUMO2 isoform (S2). **b** In actively dividing cells, SAMHD1 is targeted by CDK/cyclin-mediated phosphorylation on T592 during the G1/S transition thereby losing its antiviral activity (✗, colored in blue). A fraction of SAMHD1 (smaller circle) is SUMOylated on K595 by the action of Ubc9. However, this modification appears insufficient to fully neutralize the effects of phosphorylation and rescue restriction. Upon mitotic exit, phosphorylation is reversed by host PPP family phosphatases. Our results show that SAMHD1 harboring dephosphorylated T592 is antivirally inactive if SUMO-conjugation to K595 is prevented. They also indicate that only the fraction of SAMHD1 that harbors SUMOylated K595 and dephosphorylated T592 (✓, colored in yellow) efficiently inhibits viral infection through a dNTPase-independent mechanism.

SUMOylation (Fig. 5c). We also found that the artificial fusion of SUMO2 to a restriction-defective C-terminal truncation mutant fully restored the antiviral activity of SAMHD1. Conversely, SUMO1 had only a mild effect, despite ~50% sequence homology with its paralog[35]. These isoform-specific effects are intriguing, though they correlated with SAMHD1 having a higher affinity for SUMO2 rather than SUMO1 (Fig. 4c). Moreover, it should be noted the fraction of SUMO2/3 available for conjugation is larger than that of SUMO1, which is mostly conjugated to high-affinity targets (i.e. RanGAP1), and can further raise in response to various stimuli including viral and bacterial infection[33,44,54]. Our data point to a model where SUMOylation of K595 stimulates, while phosphorylation of T592 inhibits, the antiviral function of SAMHD1. Finding that the $T_{592}E$ mutation mitigates the restriction activity of the SAMHD1-SUMO2 chimera, although not as dramatically as in the context of WT SAMHD1, corroborates this hypothesis (Fig. 5e). This implies that phosphorylation of T592, which is important for replication fork progression and resection of collapsed forks[21], downregulates the anti-HIV-1 activity of SAMHD1 in cycling cells, irrespective of whether K595 is SUMOylated or not (Fig. 7b). Conversely, in non-cycling immune cells, where the bulk of the protein harbors a dephosphorylated T592 residue, K595 SUMOylation defines the subpopulation of restriction-competent SAMHD1 (Fig. 7b).

In a complementary approach, we assessed whether small-molecule SUMOylation inhibitors, which weakened the SAMHD1-SUMO interaction (Fig. 1e, f and Supplementary Fig. 2), might relieve the restriction activity of endogenous SAMHD1. Blocking the SUMO cascade promoted infection in contexts where SAMHD1 was antivirally active (MDMs and differentiated THP1 cells infected by HIV-1 or HIV-2ΔVpx), but not when its function was suppressed by either T592 phosphorylation (dividing THP1 cells, HIV-1 infection) or Vpx-mediated degradation (VLP-Vpx treated MDMs, infection by HIV-1; differentiated THP1 cells, infection by HIV-2) (Fig. 6). Inhibiting SUMOylation also favored the spreading of a replication-competent HIV-1 virus in cultures of primary human MDMs indicating that the observed effects are independent on the viral entry pathway. These results support the implication of SUMOylation in the antiviral response mediated by SAMDH1.

The analogous restriction phenotype of SAMHD1 variants harboring mutations that either mimic a constitutively phosphorylated T592 or impair SUMO conjugation to K595 raised the possibility that these PTMs might be connected. We clearly established that a fraction of SAMHD1 is simultaneously

phosphorylated on T592 and SUMOylated on K595 (Supplementary Fig. 9). This result agrees with the identification of the corresponding SUMO-phosphopeptide, representing ~21% of the K595 SUMOylated SAMHD1 form, by proteomic approach[47]. We also found that SUMO conjugation to K595 occurs irrespective of the presence of the phosphomimetic $T_{592}E$ or phosphoablative $T_{592}A$ mutation. Conversely, no correlation could be drawn between the lack of K595 SUMOylation and the phosphorylation status of T592, indicating that these two PTMs are, to some extent, independent events. Intriguingly, fusion of SUMO proteins to a SAMHD1 C-terminal truncation mutant lacking the CDK consensus site rescued phosphorylation of T592, probably by favoring the recruitment of SIM-containing CDK2, as shown for pRb[51]. Additional studies are required to assess whether this observation is relevant for a deeper understanding of SAMDH1 biological functions.

Several data indicate that the restriction mechanism of SAMHD1 is tightly connected to its ability to limit the dNTP supply for viral genome replication[4,8,23]. However, all the SUMOylation-defective mutants tested in this study (including $K_{595}A$, $K_{595}R$, $E_{597}Q$, SIM2m, and C-terminal truncations) were as potent as WT SAMHD1 in reducing the cellular dNTP concentrations. Along this line, SUMOylation inhibitors increased the permissiveness of myeloid cell lines to SAMHD1-sensitive viruses without altering the cellular dNTP levels. We concluded that SUMO conjugation does not influence the ability of SAMHD1 to hydrolyze dNTPs in cells. The possibility to uncouple the modulation of the dNTP pools and the antiviral function, which was previously described for the phosphomimetic $T_{592}E$[17,18,24,25] and the $C_{341}S$ and $C_{522}S$ redox-insensitive variants[55], strongly indicates that SAMHD1 possesses additional dNTPase-independent properties contributing to the viral restriction mechanism, which await identification.

The covalent attachment of SUMO to K595 might directly stimulate another enzymatic function of SAMHD1 relevant for virus restriction, i.e. the debated RNAse activity[23,26–30]. However, SUMO-conjugates represent a small fraction of SAMHD1 (~12% in transfected 293T cells (Supplementary Fig. 1), likely smaller in differentiated myeloid cells). This raises the question of how modulating the activity of a small proportion of the protein might cause the dramatic restriction defect that we observed. One can envision that, rather than acting stoichiometrically, modification of SAMHD1 by SUMO might promote a change that persists after, rather than being neutralized by, deSUMOylation, as proposed in the case of the thymine-DNA glycosylase (ref. [56] and references therein). According to this model, SUMO attachment would influence the whole population of SAMHD1, even if only a small pool of it would be modified at a given time. It is also possible that SUMOylation defines a subpopulation of SAMHD1 that is functionally and/or structurally different from the remaining pool of the protein, able for instance to interact with the incoming viral genome[27]. In turn, this association could promote SAMHD1 SUMOylation, as reported for PCNA[57] and PARP-1[58], and influence its SUMO-dependent antiviral functions. In line with this idea, we found that treatment with benzonase for removal of nucleic acids weakened the SAMHD1-SUMO2/3 association (Fig. 1d). Another major consequence of SUMOylation is the formation of new binding interfaces, leading to the notion that SUMO acts as a "molecular glue" between its substrate and a SIM-containing partner, which otherwise display weak affinity[59]. Thus, we speculate that SUMOylation of K595 might favor the recruitment of cellular and/or viral cofactors via their SIM, bringing about the formation of a complex endowed with antiviral activity. Phosphorylation of T592 might interfere with the assembly of such a complex. In an alternative scenario, SUMO as a bulky modifier might abrogate the association between SAMHD1 and an inhibitor that keeps it in an antivirally inactive state. Further studies are warranted to investigate these hypotheses.

In conclusion, our results unravel that the regulation of the antiviral function of SAMHD1 depending on the cell cycle status of the infected cell is more complex than previously anticipated and point to a scenario where phosphorylation of T592 and SUMOylation of K595 play opposite roles providing a sophisticated mechanism controlling a dNTPase-independent component of the restriction activity. These findings not only open a new perspective to uncover the enigmatic aspects of SAMHD1-mediated viral restriction, but also provide opportunities for the development of strategies aiming to selectively manipulate the immune function of SAMHD1 without affecting activities important for cell homeostasis.

## Methods

**Cells and reagents.** Human Embryonic Kidney (HEK) 293T cells were cultured in DMEM (Invitrogen). The human monocytic U937 and THP1 cell lines were grown in RPMI (Invitrogen). Media were supplemented with 10% fetal calf serum (Invitrogen) and penicillin/streptomycin (100 U/mL). U937 and THP1 cell lines were differentiated by treatment with phorbol-12-myristate-13-acetate (PMA, Sigma-Aldrich) (300 ng/mL, 24 h). All cell lines were tested mycoplasma-free (Mycoplasmacheck, GATC Biotech). Buffy coats from human healthy donors were obtained from the "Etablissement Français du Sang". Monocytes were isolated using a CD14+ selection kit (Miltenyi Biotech) and cultured 12 days in DMEM supplemented with 10% Human Serum (inactivated) to generate MDMs. Antibodies used are the following: sheep anti-SUMO1 (Enzo, 1:1000 for WB), rabbit anti-SUMO1 (ab32058, 1:1000 for PLA), rabbit anti-SUMO2/3 (ab3742, 1:1000 for WB, 1:3500 for PLA), mouse anti-SUMO2/3 (ab81371, 1:6000 for PLA), mouse anti-SAMHD1 (ab67820, 1:1000 for WB, 1:5000 for IP, 1:2500 for PLA), rabbit anti-SAMHD1 (ab177462, 1:6000 for PLA), rabbit anti-pT592-SAMHD1 (Cell Signaling #89930, 1:1000 for WB), rabbit anti-actin (Sigma-Aldrich, AA20-33, 1:2500 for WB), anti-HA HRP (Roche Clone 3F10, 1:2500 for WB), Alexa Fluor 488 anti-mouse, Alexa Fluor 594 anti-mouse (INVITROGEN, 1:800 for IF).

**Plasmid construction and mutagenesis.** pMD2.G encodes the VSVg envelope protein and psPAX2 is a second-generation HIV-1-based packaging plasmid (a gift from D. Trono). pNL4-3EnvFsGFP contains a complete HIV-1 provirus with an *env*-inactivating mutation and *EGFP* inserted in the place of the Nef-coding gene (a gift from D. Gabuzda)[60]. HIV-2 ROD9ΔEnv-GFP (WT or ΔVpx) was described previously[61]. HIV$^{NLAD8}$ is a macrophage (CCR5) tropic HIV-1 derivative of pNL4-3 containing the ADA envelope[62]. His-SUMO1, 2, and 3 were already described[63]. pLenti-puro construct expressing N-terminal HA-tagged human SAMHD1 was described previously[8] and was used as template to generate mutants using the Q5® Site-Directed Mutagenesis Kit according to the manufacturer's instructions (NEB). To obtain the SAMHD1ΔC-SUMO fusion proteins, a SalI site was inserted at position 1791 in the coding sequence of SAMHD1 (between K596 and E597). The ORF encoding SUMO1 or SUMO2 flanked by XhoI and SalI sites was amplified by PCR and then inserted by ligation into the modified vector digested by SalI. The C-terminal GG motif of SUMOs was mutated into AA to prevent conjugation. The list of oligos is provided in Supplementary Table 2. The entire coding fragment was confirmed by sequencing (GATC Biotech).

**Virus stock production, infection assay, stable cell lines.** Single-round viruses were produced by co-transfection of 293T cells using a standard calcium phosphate precipitation technique with the pNL4-3EnvFsGFP or HIV-2 ROD9ΔEnvGFP plasmids and a VSVg-expression vector (pMD2.G) at a 10:1 ratio in Ultra-CULTURE Serum-free Medium (Ozyme). Supernatants were collected 48 h post-transfection, clarified by centrifugation, filtered through 45 μm-pore size filters, and concentrated onto a 20% sucrose cushion by ultracentrifugation ($100,000 \times g$, 2 h, 6 °C) using a SW32 rotor (Beckman). HIV-1-based lentiviral particles were produced by co-transfecting 293T cells with packaging (psPAX2), VSVg-expressing (pMD2.G) and vector plasmids at a 4:1:5 ratio. Transduction experiments were performed using U937 cells at passage number 25 since the acquisition from the ATCC. Stable U937 cell lines were subject to puromycin selection (4 μg/mL, 6 days). Infection assays were conducted in a 12-well plate ($0.5 \times 10^6$ cells/well) in 3–4 technical replicates and the percentage of GFP-expressing cells was quantified after 24 or 48 h on a FORTESSA flow cytometer using a BD FACS DIVA software (BD Biosciences). Viral inocula were adjusted to yield ~40% GFP-positive PMA-treated U937 cells, corresponding to a theoretical multiplicity of infection (moi) of 0.3. MDMs were seeded in flat-bottomed 96-well plates ($10^6$ cells/well). Following incubation with GA (2 h) cells were infected with HIV$^{NLAD8}$ (10 ng/mL), and the viral p24 antigen released in the supernatant was quantified over time.

**Denaturing purification on Ni-NTA beads and immunoprecipitation assays.** 293T cells ($3 \times 10^6$ cells/10-cm dish) were transfected using a calcium phosphate

precipitation technique with plasmids encoding HA-tagged WT or mutant SAMHD1 proteins, Ubc9, and each SUMO paralog bearing an N-terminal 6-His tag or an appropriate empty plasmid. When required, cells were treated with MG132 (ON, 3 μM, Merck). Cells were lysed in ice-cold RIPA buffer (150 mM NaCl, 0,4% NaDOC, 1% IGEPAL CA-630, 50 mM Tris HCl pH 7.5, 5 mM EDTA, 10 mM NEM, 1 mM DTT, proteases cocktail inhibitors) supplemented with 1% SDS and 1% TritonX-100. Following dilution 1:5 in RIPA buffer, lysates were incubated on HA-Tag affinity matrix beads (Pierce™) (ON, RT). Alternatively, cells were lysed in buffer A (6 M guanidium-HCl, 0.1 M $Na_2HPO_4/NaH_2PO_4$, 10 mM imidazole, pH 8.0) and sonicated with a Bioruptor™ (Diagenode) (10 cycles, 45'' pulse, 20'' pause) before incubation with Nickel-Nitrilotriacetic (Ni-NTA) agarose beads (QIAGEN) (3 h, RT). Following extensive washing with decreasing concentrations of guanidium-HCl, bound proteins were eluted by boiling in Laemmli buffer supplemented with 200 mM imidazole. For native immunoprecipitation assays, PMA-treated THP1 cells ($30 \times 10^6$ cells) were lyzed in ice-cold buffer (50 mM Tris HCl pH 8.0, 280 mM NaCl, 0.5% IGEPAL, 10% glycerol, 5 mM $MgCl_2$, proteases cocktail inhibitors) supplemented with N-ethylmaleimide (NEM, 20 mM), Iodoacetamide (IAA, 5 mM) and if required, benzonase (50 U/mL) and ethidium bromide (50 μg/mL), and sonicated. Pre-cleared cell lysates were incubated with agarose beads coupled with either mouse anti-SAMHD1 antibodies or human recombinant SUMO-1 or SUMO-2 (Enzo) (ON, 4 °C). Following extensive washing in lysis buffer, bound proteins were eluted by boiling in Laemmli buffer.

**Total protein quantification and immunoblotting.** The total protein concentration was determined by Lowry's method using the DC Protein Assay Kit, according to the manufacturer's instructions (Bio-Rad) with serial dilution series of Bovine Serum Albumin (BSA, Sigma) used as calibration standard. The optical density was measured at 750 nm using a plate reader (Berthold) with MikroWin 2010 software. Proteins contained in the whole-cell lysate (WCL) or the eluates were separated on a 4–15% gradient sodium dodecylsulfate-polyacrylamide gel electrophoresis (SDS-PAGE) pre-casted gel (Bio-Rad). For $Mn^{2+}$-Phos-tag SDS-PAGE gel, the acrylamide-pendant Phos-tag ligand (50 μM, Phos-tag™ Acrylamide, FUJIFILM Wako Chemicals USA Corporation) and $MnCl_2$ (100 mM) were added to the 7% separating gel before polymerization. The Phos-tag gel was soaked in a transfer buffer containing 5 mM EDTA (10 min × 3) followed by washing in a transfer buffer without EDTA (20 min). Proteins were transferred on a nitrocellulose membrane and then detected with appropriate antibodies. Immune-complexes were revealed with HRP-conjugated secondary antibodies and enhanced chemoluminescence (Pierce™ ECL Western Blotting Substrate). Uncropped and unprocessed scans of representative blots are provided as a Source Data file.

**Immunofluorescence, in situ proximity ligation assays (Duolink), and confocal microscopy.** Cells seeded into 8-chamber culture slides (Nunc™ Lab-Tek™ II Chamber Slide™ System, ThermoFisher Scientific) ($0.4 \times 10^6$ cells/well) were fixed (4% paraformaldehyde/PBS, 15', 4 °C), permeabilized (0.1% PBS-Triton, 20', RT), quenched (125 mM glycin, 20', RT) and incubated with primary antibodies (ON, 4 °C) diluted in blocking buffer (5% BSA, 0.1% PBS-Tween20), followed by secondary antibody coupled to $Alexa_{594}$ or $Alexa_{488}$ dye (45', 30 °C). Protein-protein interactions in situ were visualized using the Duolink® in situ proximity ligation assay (PLA) system (Sigma-Aldrich) according to the manufacturer's instructions. Images were acquired using a laser-scanning confocal microscope (LSM510 Meta, Carl Zeiss) equipped with an Axiovert 200 M inverted microscope, using a Plan Apo 63/1.4-N oil immersion objective and analyzed with the imaging software Icy. The number of PLA foci per cell was scored using the same thresholding parameters across parallel samples.

**Cellular dNTPs quantification by primer extension assay.** Differentiated THP1 or U937 cell lines ($3 \times 10^6$ cells/sample) were harvested in ice-cold 65% methanol, lysed (95 °C, 3') and dried using a vacuum concentrator. The dNTP content were quantified by a single nucleotide incorporation assay as described previously[9].

**iPSC cells and iPSC-derived macrophages differentiation.** The human iPSC line OX1-61 (alternative name SFC841-03-01) used for iPSC-Macrophage differentiation in this study has been previously characterized[64]. The line was originally derived from a healthy donor recruited through the Oxford Parkinson's Disease Centre having given signed informed consent, which included the derivation of hiPSC lines from skin biopsies (Ethics Committee: National Health Service, Health Research Authority, NRES Committee South Central, Berkshire, UK (REC 10/H0505/71)). All experiments were performed in accordance with UK guidelines and regulations and as set out in the REC. The line has been deposited in the European Bank for iPS cells, EBiSC, https://cells.ebisc.org/STBCi044-A. iPSCs were cultured on Matrigel (Scientific Laboratory Supplies 354277) coated plates in mTeSR1 (Stemcell™ technologies). All iPSC stocks were QC'ed as previously described[64] and the number of passages was kept to a minimum. For iPS-MΦ differentiation[65,66], spin-embryoid bodies (EBs) were generated from iPSCs in Aggrewell™800 (Stemcell™ technologies) and differentiated for 4 days in mTeSR1 supplemented with 50 ng/mL BMP-4 (GIBCO- PHC9534), 20 ng/mL SCF

(Miltenyi Biotec) and 50 ng/mL VEGF (GIBCO- PHC9394). EBs were then transferred into a 6-well tissue culture plate in X-VIVO™-15 (Lonza) supplemented with 100 ng/mL M-CSF (Invitrogen), 25 ng/mL IL-3 (R&D), 2 mM glutamax (Invitrogen), and 0.055 mM 2-mercaptoethanol (Invitrogen). Media was changed every 5 days. Once production started, non-adherent iPS-MΦ precursors were harvested every week and terminally differentiated for one week in X-VIVO™-15, supplemented with 100 ng/mL M-CSF, 2 mM glutamax, 100 U/mL penicillin, and 100 mg/μL streptomycin.

**Statistical analyses.** Graphical representation and statistical analyses were performed using Prism7 software (GraphPad Software, San Diego, CA, USA).

**Reporting summary.** Further information on research design is available in the Nature Research Reporting Summary linked to this article.

## Data availability
Source data are provided with this paper.

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

## Acknowledgements
The authors thank H. de Thé and V.Lallemand-Breitenbach for discussion, A. Amara, and X. Carnec for critical reading of the manuscript. The authors thank M. Benkirane (IGH, Montpellier, France), N. Manel (I. Curie, Paris, France), A. Puissant (INSERM U944) and R. Hay (University of Dundee, Dundee, UK) for reagents. We are grateful to the Core facility of IRSL, Yasmine Khalil, and Renaud Batrin for technical support. The Wellcome Trust WTISSF121302 and the Oxford Martin School LC0910-004 (James Martin Stem Cell Facility Oxford). The iPS cell line used in this study was originally generated from a donor sample supplied by the Oxford Parkinson's Disease Center (OPDC) study (funded by the Monument Trust Discovery Award from Parkinson's UK, a charity registered in England and Wales (2581970) and in Scotland (SC037554), with the support of the National Institute for Health Research (NIHR) Oxford Biomedical Research Center based at Oxford University Hospitals NHS Trust and University of Oxford, and the NIHR Comprehensive Local Research Network), and was reprogrammed within StemBANCC (supported by the Innovative Medicines Initiative Joint Undertaking under grant agreement number 115439, resources of which are composed of financial contribution from the European Union's Seventh Framework Program (FP7/2007e2013) and EFPIA companies' in kind contribution). This work was supported by Sidaction (grant 2018-1-AEQ-12075 to A.Z.), Sidaction/FRM (grant VIH2016126003 to A.Z.), EU FP7 [HEALTH-2012-INNOVATION-1 'HIVINNOV'] (Grant no. 305137 to A.S. and A.Z.). C.M. was supported by fellowships from the French "Ministère de la Recherche et de l'Innovation" and Sidaction. Some experiments were performed in the laboratory of B.K., supported by NIH (grant AI136581 and AI150451 to B.K.).

## Author contributions
C.M., A.C., J.T.T., and A.Z. conceived and performed most of the experiments; N.P. performed mutagenesis and cloning; N.C., J.B., and O.S. planned and performed experiments on primary and iPS-derived macrophages; S.A.C., B.M., and B.K. provided dNTP measurements; M.P. and F.D.G. provided the SAMHD1-KO THP1 cells; G.B., L.E., and F.M.G. provided critical reagents; C.M., A.C., J.T.T., A.S., and A.Z. interpreted the data; C.M., A.C., J.T.T., and A.Z. wrote the original draft; F.M.G., P.L., and A.Z. review and edited the manuscript; A.Z. supervised the study.

## Competing interests
The authors declare no competing interests.
