## [Peer Review File · Nature Communications]

Reviewers' Comments:

Reviewer #1:

Remarks to the Author:

The authors show convincingly that SAMHD1 is SUMOylated at residue K595. Interestingly, preventing SUMOylation suppressed antiviral activity of SAMHD1 even in presence of dephosphorylated T592 residue, without affecting the dNTPase activity. This suggests that SUMOylation might be necessary and solely responsible for dNTPase-independent restriction activity of SAMHD1. The finding of a novel post-translational modification is an important advance for further understanding the regulation of SAMHD1. The manuscript contains a considerable amount of data, and experiments are carefully executed. However, the hypothesis that SUMOylation is absolutely necessary and might even override the effect of the T592 site for antiviral activity, would require more proof. In particular, only a very small portion of SAMHD1 seems to undergo SUMOylation. Do the authors believe that only a very small percentage of the protein does the job? Furthermore, how do the new results fit with previous findings that claim the antiviral activity of SAMHD1 dependent on a specific cysteine residue, also independently of T592 (PMID: 30044979). How can three variant PTMs, phosphorylation at T592, redox-active cysteine C522 and now Sumoylation at K595, all have the same effect and be independent of each other? It would help to have the findings strengthened as outlined below:

Specific points:

1) it puzzles me that SUMOylation seems to occur in both dividing and non-dividing cells, see Fig. 1B, C. Is it regulated, cell cycle dependent (as it is shown for phosphorylation at T592)? Does SUMOylation occur in G0/G1 cell cycle when T592 is mainly dephosphorylated or occur in S/G2 when T592 is phosphorylated? PMA-differentiated THP1 cells or transduced U937 cells will only partially answer this question as a fraction of SAMHD1 is still found in a phosphorylated state. This would help to clarify whether the two PTMs are dependent/independent of each other.

Furthermore, it would be very interesting to know, whether there is a relevant portion of the protein SUMOylated. This could be done by Mass-spec. This could also tell whether SUMOylation occurs concomitantly with T592 or whether both modifications are mutually exclusive. The reason why mutants might not show influence on T592 phosphorylation (in case there is an influence of SUMO) could be that SUMOylation only happens on a very small proportion of SAMHD1. Does HIV infection alter the frequency of SUMOylation on SAMHD1?

2) In some of the experiments, it will be important to show the comparison WT cells to Knockout or Knockdown of SAMHD1 in order to provide specificity:

- for the inhibitors or antibodies in the PLA experiments, e.g. Fig. 1D, Suppl. Fig. 2
- to prove the off-target band/specific band in NiTA pulldown experiments (e.g. Fig. 2B,C),
- and in particular to prove that in primary cells the inhibition of SUMOylation is dependent on SAMHD1 as inhibition of SUMOylation will have multiple effects on the viral life cycle (Fig 6C, Supp Fig 9).

3) Fig 5C and D: The intermediate phenotype of T592A/K595A is difficult to comprehend. It would be interesting to analyze the triple-mutant T592A_K595A_K596A. However, an important control for 5D would be the double-mutants T592A or E with deltaC-SUMO2

Minor points:

Fig 1B: mutants SIM2m and QQN are not explained at that point in the paper, it would be helpful to provide more explanations or reference to later

Fig S8D in comparison to Fig 5A: T592 phosphorylation of mutant Q594N differs in different cell types, 293T and U937, any explanation?

Fig 5D: what about the phosphorylation levels of these variants, presumably the deltaC variants are not phosphorylated as the cyclin motif is missing?

Suppl Fig 3: It would be interesting to discuss the murine SAMHD1 isoforms: isoform 1 does not

contain a site similar to E597, which seems to be important for SUMOylation. Moreover, more importantly, isoform 2 seem not to contain the SUMOylation site, however has been shown to be constitutively active (PMID: 2792020). How would you explain the activity of isoform 2?

Fig 5: is differing CDK binding responsible for the variant phosphorylation levels? Could SUMOylation change the ratio of cells in different cell cycle stages, could this be responsible for the variant phospho levels?

Table S1: please provide references for all listed studies

Fig 1D, Fig S6B: it seems that many PLA- dots are outside of the nucleus?

Fig S2: quantification of PLA signal would be helpful,

Fig 3 and all related infection experiments: what is the infection rate?

Fig 6B: infection rate seems extremely low. Was this the same in all repeated experiments?

Suppl Fig 9a: dATP levels might simply differ based on the size of the cells (U937 versus THP1)!! Are these cycling or PMA differentiated THP1/U937?

Reviewer #2:

Remarks to the Author:

The antiviral function of SAMHD1 in non-cycling immune cells is largely attributed to its dNTP triphosphohydrolase activity that depletes cellular dNTPs. Post translational modifications of SAMHD1 have also been considered in relation to its antiviral restriction role.

In this work, Martinat et al. investigate if SUMOylation may regulate antiviral activity in non-cycling immune cells. First, they demonstrated that SAMHD1 is conjugated by SUMOs in the nucleus of both cycling and differentiated cells. Next, they have identified which lysine residues are SUMOylated performing a SUMOylation assay in 293T cells using a considerable number of SAMHD1 mutants. In differentiated SAMHD1-deficient U937 cells they have shown that impairment of SUMOylation of lysine 595 compromises the antiviral activity of SAMHD1 but not its dNTPase function. Both K595 SUMOylation and viral restriction rely on a SIM2 motif, providing additional evidence that SAMHD1 requires K595 SUMOylation to be viral restriction competent. Interestingly, loss of 595 SUMOylation suppresses the restriction activity of SAMHD1 even in the context of the constitutively active phospho-ablative T592A SAMHD1 mutant.

This is a carefully executed study with conclusions supported by the data and will be of interest to the researches in the fields of HIV-1 restriction and SAMHD1 activities. The experiments are well described and important controls were made. The authors provide a description of the model for the regulation of SAMHD1 antiviral activity by SUMOylation and phosphorylation.

I have only a few minor points that I would like the authors to address.

Fig 1 A: Please add information about MG132 treatment in the legend

Pag 7, 189-190: In Fig S5C the catalytic-defective variants (HD/AA) is practically not express (0.1 ± 0.1), therefore the absence of effects on cellular dNTP content may not be attributed to the HD/AA mutant (Fig 3D and Fig S5E).

Reviewer #3:

Remarks to the Author:

The manuscript by Martinat et al. reports on a novel post translational modification of SAMHD1, SUMOylation. The authors show that SUMO can be added to three possible carboxy terminal residues and was later narrowed down to K569 and K622 through mutagenesis studies. The investigators then ectopically express SAMHD1 and SUMO-defective mutants thereof to ask about restriction of HIV infection. They present normalized infectivity data and claim that expression of WT SAMHD1 renders U937 cells resistant to infection. However, what is shown in Figure 3B is a

50% reduction in infectivity, which is extremely modest when compared with 90% or even higher inhibition levels seen in other studies. The narrow dynamic range of this infectivity assay is clearly a weakness here. Could this be dependent on cell type? while U937 cells are normally dividing, as they are transformed, primary macrophages are post-mitotic and exhibit a much more robust SAMHD1 phenotype (see Mlcochova PMID 29084722). Figure 3D shows a strong decrease in dATP concentrations with all the SUMO mutants, which are essentially in the same range as WT SAMHD1. Only HD/AA, the catalytic-defective mutant, fails to downregulate dATP. These data would be in agreement with the notion that restriction is independent of dNTP degradation. However, because the restriction levels are so modest, the conclusions are not very convincing. The authors also show that addition of a SUMO2 but not SUMO1 in frame to an inactive c-terminal deleted SAMHD1 restores its antiviral activity. These results are very novel and appear highly convincing. After seeing the great data in Fig. 5, I can only wonder what the dNTP levels will be like in cells transfected with the SUMO fusion proteins. I hope that such data can be made available. Very interesting indeed.

In Fig S7 and in other figures, the authors show that statistically T952 E and SIM2m are different from wild type. However, by looking at the error bars, it appears that those results are also different from the untreated U937. Therefore, dismissing T952 E and SIM2m restriction activity as being abrogated does not seem appropriate, as such activity is intermediate between 100 and 50% infection (around 75%).

In Figure 6C, the authors change to replication competent virus and also change the cell type to primary macrophages (using primary macrophages in previous figures would have strengthened the study). The inhibitors of SUMOylation have a very modest effect here although they follow the trend that was predicted, over the course of many days.

In summary, this is a novel study that has scientific merit, but many of the experiments show viral restriction levels that are low and are not quantitatively in agreement with previous studies in the literature. Given the novelty of the findings and the creation of the novel constructs, I submit that repeating select experiments with primary macrophages may settle the doubts.

We warmly thank the reviewers for their careful assessment of our work and their interesting remarks which helped us strengthen our findings and improve our manuscript.

Our point-by-point reply to their comments follows below. For clarity, our responses are highlighted in blue and changes are indicated in red in the manuscript.

REVIEWER COMMENTS

Reviewer #1 (Remarks to the Author):

The authors show convincingly that SAMHD1 is SUMOylated at residue K595. Interestingly, preventing SUMOylation suppressed antiviral activity of SAMHD1 even in presence of dephosphorylated T592 residue, without affecting the dNTPase activity. This suggests that SUMOylation might be necessary and solely responsible for dNTPase-independent restriction activity of SAMHD1. The finding of a novel post-translational modification is an important advance for further understanding the regulation of SAMHD1. The manuscript contains a considerable amount of data, and experiments are carefully executed. However, the hypothesis that SUMOylation is absolutely necessary and might even override the effect of the T592 site for antiviral activity, would require more proof. In particular, only a very small portion of SAMHD1 seems to undergo SUMOylation. Do the authors believe that only a very small percentage of the protein does the job?

We are grateful to reviewer#1 for pointing out that our study is relevant and of high quality. We also thank him/her for pointing out that our conclusions on the role of K595 SUMOylation and T592 phosphorylation in the regulation of the antiviral activity of SAMHD1 was not clear enough.

We do not believe that SUMOylation is solely responsible for the activation of SAMHD1 antiviral activity in non-cycling immune cells. However, we find that dephosphorylated T592 alone is insufficient to render SAMHD1 restriction competent in a context where K595 SUMOylation is inhibited (**New Fig. 5C**). Conversely, the phosphomimetic T₅₉₂E mutation mitigates the restriction activity of a SAMHD1-SUMO2 chimera (**New Fig. 5F**). Overall, these observations imply that optimal antiviral activity is witnessed when SAMHD1 harbors both SUMOylated K595 and unphosphorylated T592, which occurs in non-dividing cells. In cycling cells phosphorylation of T592 antagonizes the effect of K595 SUMOylation, relieving the SAMHD1-mediated restriction. We have clarified these points in the revised version of our manuscript (**Page 13, Lines 410-414**).

As the reviewer points out, only a small fraction of SAMHD1 is SUMOylated (**see also answer to point 1.b**), which is the case for most SUMO substrates¹. It is possible that SUMOylation defines a subpopulation of SAMHD1 that is functionally and/or structurally different from the remaining pool of the protein, able for instance to interact with the incoming viral genome and trigger the formation of a complex endowed with antiviral activity. One can also envision that, rather than acting stoichiometrically, modification of SAMHD1 by SUMO

might promote a change that persists after, rather than being neutralized by deSUMOylation, as proposed in the case of the thymine-DNA glycosylase² (for a review¹ and references therein). According to this model, SUMO attachment would influence the whole population of SAMHD1, even if only a small pool of it would be modified at a given time. These hypotheses have been included in the discussion of the revised version of the manuscript (**Page 14, Lines 459-469**).

Furthermore, how do the new results fit with previous findings that claim the antiviral activity of SAMHD1 dependent on a specific cysteine residue, also independently of T592 (PMID: 30044979). How can three variant PTMs, phosphorylation at T592, redox-active cysteine C522 and now Sumoylation at K595, all have the same effect and be independent of each other?

As indicated by the reviewer, Wang et al. reported that mutation of the redox-sensitive C341 or C522 residues abolished the restriction activity of SAMHD1, even in a context where T592 was dephosphorylated³. In agreement with this observation, we clearly demonstrate that dephosphorylated T592 is required, but not sufficient to render SAMHD1 antivirally active. It is tempting to speculate that the oxidation state of SAMHD1 might influence its modification by SUMO, as previously shown for PML/Trim19⁴, a question that remains open for future studies.

On the other hand, we have studied in detail the functional connection between T592 phosphorylation and K595 SUMOylation in the regulation of the restriction activity of SAMHD1. Although we found that a fraction of SAMHD1 bears concomitantly phosphorylated T592 and SUMOylated K595 (**New Fig. S9B**), we were unable to establish a hierarchical relationship between these post-translational modifications (**Fig. S10**), implying that they are at least in part independent processes. These data are presented in the revised version of our manuscript (**Pages 8-9, Lines 257-267**).

It would help to have the findings strengthened as outlined below:

Specific points:

1.a) it puzzles me that SUMOylation seems to occur in both dividing and non-dividing cells, see Fig. 1B, C. Is it regulated, cell cycle dependent (as it is shown for phosphorylation at T592)?

Does SUMOylation occur in G0/G1 cell cycle when T592 is mainly dephosphorylated or occur in S/G2 when T592 is phosphorylated? PMA-differentiated THP1 cells or transduced U937 cells will only partially answer this question as a fraction of SAMHD1 is still found in a phosphorylated state. This would help to clarify whether the two PTMs are dependent/independent of each other.

As indicated by the reviewer, our results show that SAMHD1 is a SUMO target in both dividing and non-dividing cells. The novel data showing that **i)** high-molecular weight bands reactive to anti-SUMO isoform specific antibodies are enriched upon immunoprecipitation of endogenous SAMHD1 from differentiated THP1 cells (**New Fig. 1D**) and that **ii)** SAMHD1 and SUMO2/3 interact in the nucleus of primary human MDMs (**New Figs. 1E, 1F and S2A**) further indicate SAMHD1 SUMOylation is not influenced by the cell division status, although site-specific variations depending on the cellular context cannot be excluded (These new data are described at **Pages 4-5, Lines 103-122**).

As requested by the reviewer, we also undertook cell cycle synchronization experiments to study the dynamics of SAMHD1 SUMOylation along the cell division cycle phases. We transduced HeLa cells stably expressing His-SUMO2 (a kind gift of R. Hay, University of Dundee, UK), or parental HeLa cells, with a lentivector encoding HA-SAMHD1 **RKR** mutant (where K469 and K622 are simultaneously mutated to R and thus only K595 contributes to the SUMOylated fraction). Cells were either left asynchronously growing (NS) or synchronized by double-thymidine block and released for 2 or 11 hours (corresponding to S- and G1-phase synchronized cells, respectively⁵). Following denaturing lysis (1% SDS-containing buffer), samples were diluted in SDS-less buffer (1:5) and then incubated on HA-matrix beads. Proteins from both the input and the eluates were analyzed with anti-pT592, anti-SUMO2/3 or anti-HA antibody to detect the phosphorylated, SUMOylated or total SAMHD1 species, respectively.

As shown in the figure below, T592 phosphorylated SAMHD1 species were detected at 2, but not 11, hours post-release (hpr), in agreement with previous observations⁵. T592 phosphorylation was similarly undetectable in the NS conditions, likely because a large fraction of the cell population is in the G1 phase. High-molecular weight SAMHD1 species reactive to anti-SUMO isoform specific antibodies were also enriched in the 2 hpr sample as compared to the 11 hpr and NS conditions. Despite comparable expression of SAMHD1 **RKR** mutant in all samples, SUMOylated forms were undetectable in control HeLa cells (ctr). These results indicate that SUMOylation of K595, like phosphorylation of T592, is present during the S phase, consistent with the existence of a subpopulation of co-modified SAMHD1 (**New Fig. S9C**, and answer to point 1.b below). Having established that SAMHD1 is SUMOylated in non-dividing cells (**Fig. 1**), the absence of SUMO-conjugates in G0/G1 cells surprised us. We hypothesize that the SUMO-modified fraction of SAMHD1 in these samples might fall below the detection limit or that cell type-specific differences might exist. Further investigations are required to test these ideas and fully characterize the SUMOylation status of SAMHD1 along the different phases of the cell cycle.

1.b) Furthermore, it would be very interesting to know, whether there is a relevant portion of the protein SUMOylated. This could be done by Mass-spec. This could also tell whether SUMOylation occurs concomitantly with T592 or whether both modifications are mutually exclusive. The reason why mutants might not show influence on T592 phosphorylation (in case there is an influence of SUMO) could be that SUMOylation only happens on a very small proportion of SAMHD1.

Determining the SUMO-conjugated fraction of a protein by MS is challenging given **i)** the large C-terminal SUMO remnant following tryptic digestion, which complicates the interpretation of the MS fragment-ion⁶, **ii)** the low abundance and dynamic nature of this modification, **iii)** the simultaneous conjugation of SUMO to multiple sites, which makes it currently impossible to know whether different SUMOylated peptides derive from the same molecule or not. To overcome these issues, and identify SUMO substrates as well as the modified sites, typical experimental strategies rely on denaturing cell lysis and enrichment of ectopically expressed His-tagged SUMO mutants, which leave a shorter signature peptide upon protease digestion. Using these approaches, several groups found that endogenous SAMHD1 is a SUMO target (**Table 1**). Hendricks et al.⁷ ranked SAMHD1 among the 8% top SUMOylated hits (507/6747). They also found that 48% (22/46) Lysine residues of SAMHD1 are SUMOylatable and defined the contribution of each of them to the overall modification profile of the protein (a parameter called fractional intensity) (**Table 1**).

In the SUMO field, it is well established that the SUMOylated forms represent only a small pool for most substrates¹. However, quantitative data are available for a limited number of well-known SUMO target proteins (including CtIP⁸, TOP1^{9,10}, TDG², Stat1¹¹; RanGAP1¹²) and are generally obtained by densitometric analysis of bands corresponding to the unmodified

and SUMOylated species with the ImageJ software. Using this strategy, we established that the average ratio of SUMOylated *versus* total SAMHD1 is ~12% (**New Fig. S1, page 4, lines 89-90**). Similarly, we determined that ~6% of T592 phosphorylated SAMHD1 fraction is concomitantly SUMOylated (**New Fig. S9B, pages 8-9, lines 257-267**). These findings agree with the identification of the corresponding SUMO-phosphopeptide by Hendriks et al.⁷, which represented ~21% of the K595 SUMOylated form. Overall, these observations indirectly point to the existence of various SAMHD1 proteoforms, bearing either phosphorylated T592 or SUMOylated K595 or both, which might contribute to the spatial and timely regulation of distinct biological functions.

1.c) Does HIV infection alter the frequency of SUMOylation on SAMHD1?

During our investigations, we tested whether nucleic acids binding might influence the SAMHD1/SUMO interaction. Thus, differentiated THP1 cells were lysed and SAMHD1 immunoprecipitated in the presence, or the absence, of benzonase treatment followed by immunoblotting with anti-SUMO isoform specific antibodies. THP1 cells where the *SAMHD1* gene was knocked down using the CRISPR/Cas9 technology served as negative control. We detected high-molecular weight forms reactive to anti-SUMO1 or anti-SUMO2/3 antibodies in WT, but not in SAMHD1-KO THP1 cells. Incubation with benzonase reduced the amounts of SUMO-conjugates associated to SAMHD1-coated beads, indicating that nucleic acids might favor the SAMHD1-SUMO interaction. These results are presented in **New Fig. 1D** and described in the “Results” section (**Pages 4, Lines 103-110**).

We also analyzed the SAMHD1-SUMO interaction in differentiated THP1 cells challenged with a VSVg/HIV-1 GFP virus (moi 2.5 or 5) using the Proximity Ligation Assay (PLA). Cells were fixed 4 or 16 hours post infection (hpi) and, next, processed for PLA using anti-SAMHD1 and anti-SUMO2/3. As shown in the figure below, the SAMHD1-SUMO2/3 proximity signal, normalized on the bases of SAMHD1 expression levels, increased in infected cells, as compared to the non-infected control sample (NI, arbitrarily set to 1). These results suggest that HIV-1 infection might stimulate SAMHD1 SUMOylation, a conclusion that needs to be confirmed by further investigations.

2) In some of the experiments, it will be important to show the comparison WT cells to Knockout or Knockdown of SAMHD1 in order to provide specificity:

2.a) for the inhibitors or antibodies in the PLA experiments, e.g. Fig. 1D, Suppl. Fig. 2

As required by the reviewer, we added the appropriate controls to validate the specificity of the SAMHD1-SUMO2 PLA signal.

1. A panel showing the PLA signal in PMA-treated U937 cells stained with anti-SAMHD1 and anti-SUMO2/3 antibodies, with the corresponding quantification, has been included in **New Fig. S7B and S7C** of the revised version of the manuscript (**Page 8, Line 227**).

2. We also performed new PLA experiments in primary human MDMs which were incubated in the presence of VLP harboring Vpx, or not, and then treated with ginkgoid acid (GA) or left untreated. The SAMHD1-SUMO2/3 interaction was readily detected in the nucleus of untreated MDMs but was virtually abolished upon Vpx-mediated degradation of SAMHD1, which validates the specificity of the SAMHD1-SUMO2/3 proximity labeling. Incubation with GA also lowered the frequency of the SAMHD1-SUMO2/3 PLA signal, but had no consequences if SAMHD1 had previously been degraded. These results showing that SAMHD1 and SUMO2/3 interact in the nucleus of MDM are included in **New Figs. 1E, 1F** and **S2A** and described in the “Results” section of the revised version of the manuscript (**Page 5, Lines 111-122**).

2.b) to prove the off-target band/specific band in Ni-NTA pulldown experiments (e.g. Fig. 2B,C)

Please find below a figure showing that a ~70 kDa size band is visualized with an anti-HA antibody following incubation of Ni-NTA beads with the lysate of 293T cells transfected with a plasmid encoding HA-SAMHD1, but not the control empty vector (ctr) (compare lanes 1 and 2, and lanes 4 and 5). Note that slow migrating bands (>100 kDa) are detected only when HA-SAMHD1 and His-SUMO2 are co-expressed (lanes 3 and 6).

2.c) and in particular to prove that in primary cells the inhibition of SUMOylation is dependent on SAMHD1 as inhibition of SUMOylation will have multiple effects on the viral life cycle (Fig 6C, Supp Fig 9).

We performed new infection experiments where primary human MDMs generated from healthy donors (including those of New Figs. 1E, 1F and S2A) were incubated with VLP harboring Vpx, to downregulate SAMHD1 expression, or not. Next, cells were treated with ginkgolic acid (GA) or vehicle before being challenged with the replication-competent macrophage-tropic AD8 strain HIV-1. Consistent with our early results on THP1 cells, inhibiting the SUMO cascade favored viral spread in SAMHD1-expressing cells, although to a various degree depending on the donor, but had no additional effects on viral replication if SAMHD1 had been previously degraded. These results are presented in **New Fig. 6C** and described in the “Results” section of the revised version of the manuscript (**Page 11, Lines 360-361**).

3) Fig 5C and D: The intermediate phenotype of T592A/K595A is difficult to comprehend. It would be interesting to analyze the triple-mutant T592A_K595A_K596A. However, an important control for 5D would be the double-mutants T592A or E with deltaC-SUMO2

As requested by the reviewer, we generated the triple SAMHD1 T₅₉₂A_K₅₉₅₋₅₉₆R mutant and performed the U937-based restriction assay. As shown in **New Fig. S11**, the triple mutant was slightly less potent than the T₅₉₂A_K₅₉₅R variant in inhibiting HIV-1 infection. We speculate that SAMHD1 harboring the conservative K-to-R substitution at position 595 and 596 might still be able to recruit the SUMO-charged Ubc9, therefore allowing the modification of alternative nearby site(s). Indeed, five additional SUMOylatable K residues lie near the SIM and the SUMO-consensus motif centered on K595 (**Table 1** and **New Fig. S11C**). To test further this idea, we generated a SAMHD1 T₅₉₂A variant where SIM2 was inactivated and found that it mirrored the loss-of-restriction phenotype of the T₅₉₂A_E₅₉₇Q mutant (**New Fig. 5C**). These data, which support the requirement of K595 SUMOylation for efficient viral restriction by SAMHD1, are described in the “Results” section (**Page10, Lines 297-306**).

To bring additional evidence for the implication of K595 SUMOylation in the SAMHD1-mediated antiviral mechanism, we also assessed the consequences of progressively truncating the C-terminal region of SAMHD1 on the restriction activity. Deletion of the 26 C-terminal amino acids (yielding HA-SAMHD1 Δ C₆₀₀) altered neither the integrity of the SUMO-consensus motif centered on K595 (**New Fig. 5D**), nor antiviral activity (**New Fig. 5E**). Conversely, SAMHD1 truncation mutants harboring partial (HA-SAMHD1 Δ C₅₉₇) or complete deletion (HA-SAMHD1 Δ C₅₉₅) of the SUMOylation site (**New Fig. 5D**) were inactive against HIV-1 (**New Fig. 5E**), as previously shown^{23,49}.

Finally, as requested by reviewer#1, we introduced the phospho-ablative T₅₉₂A or phosphomimetic T₅₉₂E change within the SAMHD1 Δ C₅₉₇-SUMO2 chimera and tested the antiviral function of the resulting mutants. Replacing T592 with A did not modify the anti-HIV-1 activity of the SAMHD1-SUMO2 fusion protein, while the T₅₉₂E mutation relieved its restriction activity, yet not as dramatically as in the context of WT SAMHD1 (**New Fig. 5E**). Notably, all the SAMHD1 constructs lowered the dNTP levels to the same extent as WT SAMHD1 (**New Figs. 5F and S12A**). These novel data, which strongly suggest that SUMOylation of K595 and phosphorylation of T592 play opposite roles in the regulation of a dNTPase-independent antiviral activity of SAMHD1, are described in **Page 10, lines 307-314 and Lines 321-332**.

Minor points:

Fig 1B: mutants SIM2m and QQN are not explained at that point in the paper, it would be helpful to provide more explanations or reference to later

As suggested by the reviewer we included a reference to later in the text (**Page 4, Line 96**).

Fig S8D in comparison to Fig 5A: T592 phosphorylation of mutant Q594N differs in different cell types, 293T and U937, any explanation?

Our observations showing that mutation of Q594 dramatically reduces the T592 phosphorylation levels in differentiated U937 cells agrees with the results of Monit et al.¹³. The milder decrease of T592 phosphorylation observed in dividing 293T cells might be the consequence of cell type-specific regulatory mechanisms, or higher protein expression levels.

Fig 5D: what about the phosphorylation levels of these variants, presumably the deltaC variants are not phosphorylated as the cyclin motif is missing?

As requested by the reviewer, we analyzed the phosphorylation status of T592 for the various SAMHD1 C-terminal deletion mutants. Expectedly, the band corresponding to T592 phosphorylated SAMHD1 become undetectable upon deletion of residues 595-626 (Δ C₅₉₅), which compromises the integrity of the CDK-consensus motif (**New Fig. S12B**). Conversely, SAMHD1 variants where the CDK-consensus motif is intact (Δ C₆₀₀ and Δ C₅₉₇), displayed low but detectable levels of T592 phosphorylation even if the C-terminal cyclin binding site (L620/F621)¹⁴ is missing (**New Fig. S12B**). A plausible explanation lies in the presence of a second site spanning R451 and L453 that mediates cyclin A2 recruitment¹⁵. Surprisingly, T592 phosphorylation increased ~4- and 2-fold upon fusion of SUMO1 or SUMO2, respectively, to SAMHD1 Δ C₅₉₇. These results seem somewhat is contradiction with the

observation that SUMOylation-deficient mutants (E597Q, SIM2m) retain WT phosphorylation levels (**New Figs. 5A, S8B and S8C**). Intriguingly, a recent report showed that SUMOylation of pRb favors its phosphorylation by promoting the recruitment of SIM-containing CDK2¹⁶. Further investigations are warranted to assess whether SUMOylation of K595 plays a similar role for SAMHD1, or whether increased phosphorylation is an indirect consequence of the artificial fusion of SUMO. These results are now described in the text of the “Results” (**Page 10, Lines 321-332**) and “Discussion” sessions (**Pages 13-14, Lines 434-445**).

Suppl Fig 3: It would be interesting to discuss the murine SAMHD1 isoforms: isoform 1 does not contain a site similar to E597, which seems to be important for SUMOylation. Moreover, more importantly, isoform 2 seem not to contain the SUMOylation site, however has been shown to be constitutively active (PMID: 2792020). How would you explain the activity of isoform 2?

Although human and murine SAMHD1 isoforms are highly similar in sequence (>70% sequence identity), structure, and function, published data suggest that the mechanisms regulating their dNTPase and antiviral activities might be at least in part different. For instance, we found that deletion of the C-terminal region encompassing aa 595-626 (which compromises the integrity of the SUMOylation site) abolishes the antiviral activity of human SAMHD1 (**New Figs. 5E and 5F, Page 10, Lines 307-314**), as previously shown^{17,18}. Conversely, a murine SAMHD1 C-terminal truncation mutant efficiently inhibits HIV-1 infection¹⁹. Recent structural studies also reveal differential requirements for the SAM domain: it appears essential for both the dNTPase and restriction functions of murine SAMHD1 isoform 2²⁰, while it is dispensable in the case of the human protein^{18,21}. In the same line, human and murine SAMHD1 variants seems to have both overlapping and distinct binding partners^{5,22}. By analyzing the sequence of murine SAMHD1 isoforms with the JASSA bioinformatic tools we identified a putative inverted SUMOylation motif overlapping with a SIM (⁹³**EKKVLDI**⁹⁹) within the common SAM domain (figure A below). By mapping the position of these sites on the crystal structure of the murine SAMHD1 tetramer, we found that they are surface exposed (figure B below; the SAM domain of one protomer is colored in dark grey; PDB: 6BRG) raising the possibility that regulation of the antiviral activity by SUMOylation might be a conserved mechanism.

Fig 5: is differing CDK binding responsible for the variant phosphorylation levels? Could SUMOylation change the ratio of cells in different cell cycle stages, could this be responsible for the variant phospho levels?

In agreement with our findings, Yan and al. have previously shown that simultaneous substitution of A for Q594, K595 and K596 impaired cyclin A/CDK2 complex-mediated T592 phosphorylation *in vitro*¹⁴. The corresponding reference has been added in the text of the manuscript (**Page 9, Line 281**). Besides, we did detect neither major alteration in the distribution of U937 cells stably expressing SAMHD1 mutants along the different stages of the division cycle (see figure below) nor significant difference in the dNTP levels (**Figs. 3D and S5E**) suggesting that cell metabolism was not altered. Collectively, these observations indicate that inactivation of the CDK consensus motif most likely underlies impaired phosphorylation of T592 for the SAMHD1 variants.

Table S1: please provide references for all listed studies

As requested by reviewer#1 the references have been added.

Fig 1D, Fig S6B: it seems that many PLA- dots are outside of the nucleus?

The reviewer is right. Occasionally PLA foci are found in the cytoplasm, where a fraction of SAMHD1 resides²⁶ and might interact with SUMO proteins. However, low levels of cytoplasmic SAMHD1-SUMO2/3 PLA labelling is also visualized in VLP-Vpx treated MDMs, indicative of background signal (**New Fig. 1D** of the revised version).

Fig S2: quantification of PLA signal would be helpful,

As requested by the reviewer, these quantitative data were included in **Fig. S2** in the revised version of the manuscript. Of note, the use of anti-SUMO isoform specific antibodies does not allow a straightforward comparison between the intensity of SAMHD1-SUMO1 and SAMHD1-SUMO2/3 PLA signals.

Fig 3 and all related infection experiments: what is the infection rate?

As stated in the material and methods section (**Page 16, Lines 535-537**) viral inocula were adjusted to yield ~40% GFP-positive PMA-treated U937 cells (0.5×10^6 cells/well in 12-well plates), corresponding to a theoretical multiplicity of infection (moi) of 0.3.

Fig 6B: infection rate seems extremely low. Was this the same in all repeated experiments?

Differentiated THP1 were infected with VSVg-pseudotyped HIV-2 or HIV-2 Δ Vpx expressing GFP as reporter at a theoretical moi of 0.3, as established by titration on differentiated U937 cells. Similar infection rates were obtained in two independent experiments each performed in 3 technical replicates (**Fig. 6B**).

Suppl Fig 9a: dATP levels might simply differ based on the size of the cells (U937 versus THP1)!! Are these cycling or PMA differentiated THP1/U937?

Quantification of the dNTP pools was performed using normalized number of PMA-treated THP1 or U937 cells ($\sim 3 \times 10^6$ cells per sample). This information has been included in the material and methods section (**Page 18, Line 592**).

Reviewer #2 (Remarks to the Author):

The antiviral function of SAMHD1 in non-cycling immune cells is largely attributed to its dNTP triphosphohydrolase activity that depletes cellular dNTPs. Post translational modifications of SAMHD1 have also been considered in relation to its antiviral restriction role.

In this work, Martinat et al. investigate if SUMOylation may regulate antiviral activity in non-cycling immune cells. First, they demonstrated that SAMHD1 is conjugated by SUMOs in the nucleus of both cycling and differentiated cells. Next, they have identified which lysine residues are SUMOylated performing a SUMOylation assay in 293T cells using a considerable number of SAMHD1 mutants. In differentiated SAMHD1-deficient U937 cells they have shown that impairment of SUMOylation of lysine 595 compromises the antiviral activity of SAMHD1 but not its dNTPase function. Both K595 SUMOylation and viral restriction rely on a SIM2 motif, providing additional evidence that SAMHD1 requires K595 SUMOylation to be viral restriction competent. Interestingly, loss of 595 SUMOylation suppresses the restriction activity of SAMHD1 even in the context of the constitutively active phospho-ablative T592A SAMHD1 mutant.

This is a carefully executed study with conclusions supported by the data and will be of interest to the researches in the fields of HIV-1 restriction and SAMHD1 activities. The experiments are well described and important controls were made. The authors provide a description of the model for the regulation of SAMHD1 antiviral activity by SUMOylation and phosphorylation.

I have only a few minor points that I would like the authors to address.

We thank reviewer#2 for pointing out the quality of our study and the relevance of our observations towards a better understanding of the mechanisms regulating SAMHD1 functions. the activity of SAMHD1.

Fig 1 A: Please add information about MG132 treatment in the legend.

As requested by the reviewer, this information has been included in the new version of the manuscript (Page 16, Line 545) as well as figure 1A legend.

Pag 7, 189-190: In Fig S5C the catalytic-defective variants (HD/AA) is practically not express (0.1 ± 0.1), therefore the absence of effects on cellular dNTP content may not be attributed to the HD/AA mutant (Fig 3D and Fig S5E).

We agree with the reviewer's comment and removed the data relative to the HD/AA SAMHD1 variant. SAMHD1 T592E mutant was used throughout the experiments as loss-of-restriction control.

Reviewer #3 (Remarks to the Author):

The manuscript by Martinat et al. reports on a novel post translational modification of SAMHD1, SUMOylation. The authors show that SUMO can be added to three possible carboxy terminal residues and was later narrowed down to K569 and K622 through mutagenesis studies. The investigators then ectopically express SAMHD1 and SUMO-defective mutants thereof to ask about restriction of HIV infection. They present normalized infectivity data and claim that expression of WT SAMHD1 renders U937 cells resistant to infection. However, what is shown in Figure 3B is a 50% reduction in infectivity, which is extremely modest when compared with 90% or even higher inhibition levels seen in other

studies. The narrow dynamic range of this infectivity assay is clearly a weakness here. Could this be dependent on cell type? while U937 cells are normally dividing, as they are transformed, primary macrophages are post-mitotic and exhibit a much more robust SAMHD1 phenotype (see Mlcochova PMID 29084722).

1) Figure 3D shows a strong decrease in dATP concentrations with all the SUMO mutants, which are essentially in the same range as WT SAMHD1. Only HD/AA, the catalytic-defective mutant, fails to downregulate dATP. These data would be in agreement with the notion that restriction is independent of dNTP degradation. However, because the restriction levels are so modest, the conclusions are not very convincing.

During our studies we observed a variation in the magnitude of the restriction effect of SAMHD1 stably expressed in U937 cells ranging from 2.5- to 5-fold. Notable similar variations in the levels of inhibition are found in published studies (see for instance²⁷⁻³¹). Nevertheless, the monocytic U937 cell line, which lacks detectable expression of endogenous SAMHD1, represents a privileged experimental model for these analyses so far.

Conversely, primary MDMs are not the system of choice to characterize the restriction phenotype of SAMHD1 mutants. Indeed, this would require prior degradation of the endogenous protein to allow the efficient expression of SAMHD1 variants by lentiviral transduction, which is technically very challenging. Moreover, SAMHD1-mediated antiviral function seems to require the formation oligomers, raising the possibility that the endogenous protein might interfere with the function of the studied mutant.

To strengthen the relevance of the loss-of-restriction phenotype observed when K595 SUMOylation is impaired, we repeated all the restriction assays using the same batch of U937 cells which were transduced at passage 25 since the acquisition from the ATCC repository (as stated in the material and methods of the revised version of the manuscript, **(page 16, Lines 530-531)**). Beside the stronger inhibitory effect of WT SAMHD1 (5- to 6-fold restriction as compared to parental U937 cells), the new results fully agree with the trend of our previous data, as shown below.

2) The authors also show that addition of a SUMO2 but not SUMO1 in frame to an inactive c-terminal deleted SAMHD1 restores its antiviral activity. These results are very novel and appear highly convincing. After seeing the great data in Fig. 5, I can only wonder what the dNTP levels will be like in cells transfected with the SUMO fusion proteins. I hope that such data can be made available. Very interesting indeed.

We thank reviewer#3 for acknowledging the novelty and quality of our findings. As requested, we quantified the dNTP levels in PMA-treated U937 cells expressing the SAMHD1-SUMO chimeras and found them in the same range as those of WT SAMHD1-expressing cells. These data are included in **New Fig. 5F** and **S12A** of the revised version of the manuscript and described in the “Results” section (**Page 10, Lines 321-332**).

3) In Fig S7 and in other figures, the authors show that statistically T952 E and SIM2m are different from wild type. However, by looking at the error bars, it appears that those results are also different from the untreated U937. Therefore, dismissing T952 E and SIM2m restriction activity as being abrogated does not seem appropriate, as such activity is intermediate between 100 and 50% infection (around 75%).

As indicated above (point 1), we repeated the restriction experiment and established that mutation of SIM2 results in a ~3-fold reduction of the restriction activity as compared to WT SAMHD1 (**New Figs. 4E**). A similar loss-of-restriction phenotype was observed when SIM2 was inactivated in the context of the constitutively active T₅₉₂A variant (yielding SAMHD1 T₅₉₂A_SIM2m) (**New Fig. 5C, page 10, lines 297-306**).

4) In Figure 6C, the authors change to replication competent virus and also change the cell type to primary macrophages (using primary macrophages in previous figures would have strengthened the study). The inhibitors of SUMOylation have a very modest effect here although they follow the trend that was predicted, over the course of many days.

As indicated above (answer to reviewer#1's point 2.c, pages 8-9), we performed new infection experiments where primary human MDMs were incubated with VLP harboring Vpx, to downregulate SAMHD1 expression, or not. Next, cells were treated with ginkgolic acid (GA) or vehicle before being challenged with the replication-competent macrophage-tropic AD8 strain HIV-1. Our new results agree with the data presented in the previous version of the manuscript, and show that inhibiting the SUMO cascade favors viral spread in SAMHD1-expressing cells, although to a various degree depending on the donor, but had no additional effects on viral replication if SAMHD1 had been previously degraded. These results are presented in **New Fig. 6C** and described in the "Results" section of the revised version of the manuscript (**Page 11, Lines 356-361**).

In summary, this is a novel study that has scientific merit, but many of the experiments show viral restriction levels that are low are not quantitatively in agreement with previous studies in the literature. Given the novelty of the findings and the creation of the novel constructs, I submit that repeating select experiments with primary macrophages may settle the doubts.

Bibliography

1. Johnson, E. S. Protein modification by SUMO. *Annu. Rev. Biochem.* **73**, 355–82 (2004).
2. Hardeland, U. & Steinacher, R. Modification of the human thymine-DNA glycosylase by ubiquitin-like proteins facilitates enzymatic turnover. *EMBO J.* **21**, 1456–1464 (2002).
3. Wang, Z. *et al.* Functionality of Redox-Active Cysteines Is Required for Restriction of Retroviral Replication by SAMHD1. *Cell Rep.* **24**, 815–823 (2018).
4. Sahin, U., de Thé, H. & Lallemand-Breitenbach, V. PML nuclear bodies: Assembly and oxidative stress-sensitive sumoylation. *Nucleus* **5**, 499–507 (2014).
5. Schott, K. *et al.* Dephosphorylation of the HIV-1 restriction factor SAMHD1 is mediated by PP2A-B55 α holoenzymes during mitotic exit. *Nat. Commun.* **9**, 2227 (2018).
6. Zhang, Y., Li, Y., Tang, B. & Zhang, C. Y. The strategies for identification and quantification of SUMOylation. *Chem. Commun.* **53**, 6989–6998 (2017).
7. Hendriks, I. A. *et al.* Site-specific mapping of the human SUMO proteome reveals co-modification with phosphorylation. *Nat. Struct. Mol. Biol.* **24**, 325–336 (2017).
8. Locke, A. J. *et al.* SUMOylation mediates CtIP's functions in DNA end resection and replication fork protection. *Nucleic Acids Res.* **49**, 928–953 (2021).
9. Desai, S. D., Li, T. K., Rodriguez-Bauman, A., Liu, L. F. & Rubin, E. H. Ubiquitin/26s proteasome-mediated degradation of topoisomerase I as a resistance mechanism to

- camptothecin in tumor cells. *Cancer Res.* **61**, 5926–5932 (2001).
10. Desai, S. D., Liu, L. F., Vazquez-Abad, D. & D'Arpa, P. Ubiquitin-dependent destruction of topoisomerase I is stimulated by the antitumor drug camptothecin. *J. Biol. Chem.* **272**, 24159–24164 (1997).
 11. Song, L., Bhattacharya, S., Yunus, A. A., Lima, C. D. & Schindler, C. Stat1 and SUMO modification. *Blood* **108**, 3237–3244 (2006).
 12. Matunis, M. J., Wu, J. & Blobel, G. SUMO-1 modification and its role in targeting the Ran GTPase-activating protein, RanGAP1, to the nuclear pore complex. *J. Cell Biol.* **140**, 499–509 (1998).
 13. Monit, C. *et al.* Positive selection in dNTPase SAMHD1 throughout mammalian evolution. *Proc. Natl. Acad. Sci.* **116**, 201908755 (2019).
 14. Yan, J. *et al.* CyclinA2-Cyclin-dependent Kinase Regulates SAMHD1 Protein Phosphohydrolase Domain. *J. Biol. Chem.* **290**, 13279–13292 (2015).
 15. Gelais, C. S. *et al.* A putative cyclin-binding motif in human SAMHD1 contributes to protein phosphorylation, localization, and stability. *J. Biol. Chem.* **291**, 26332–26342 (2016).
 16. Meng, F., Qian, J., Yue, H., Li, X. & Xue, K. SUMOylation of Rb enhances its binding with CDK2 and phosphorylation at early G1 phase. *Cell Cycle* **15**, 1724–1732 (2016).
 17. Yan, J. *et al.* Tetramerization of SAMHD1 is required for biological activity and inhibition of HIV infection. *J. Biol. Chem.* **288**, 10406–10417 (2013).
 18. Arnold, L. H. *et al.* Phospho-dependent Regulation of SAMHD1 Oligomerisation Couples Catalysis and Restriction. *PLoS Pathog.* **11**, 1–30 (2015).
 19. Bloch, N. *et al.* A highly active isoform of lentivirus restriction factor SAMHD1 in mouse. *J. Biol. Chem.* **292**, 1068–1080 (2017).
 20. Buzovetsky, O. *et al.* The SAM domain of mouse SAMHD1 is critical for its activation and regulation. *Nat. Commun.* **9**, SUPP DATA (2018).
 21. White, T. E. *et al.* Contribution of SAM and HD domains to retroviral restriction mediated by human SAMHD1. *Virology* **436**, 81–90 (2013).
 22. St Gelais, C. *et al.* Identification of Cellular Proteins Interacting with the Retroviral Restriction Factor SAMHD1. *J. Virol.* **88**, 5834–5844 (2014).
 23. Laguette, N. *et al.* Evolutionary and functional analyses of the interaction between the myeloid restriction factor SAMHD1 and the lentiviral Vpx protein. *Cell Host Microbe* **11**, 205–217 (2012).
 24. Lim, E. S. *et al.* The ability of primate lentiviruses to degrade the monocyte restriction factor SAMHD1 preceded the birth of the viral accessory protein Vpx. *Cell Host Microbe* **11**, 194–204 (2012).
 25. Fregoso, O. I. *et al.* Evolutionary Toggling of Vpx/Vpr Specificity Results in Divergent Recognition of the Restriction Factor SAMHD1. *PLoS Pathog.* **9**, e1003496 (2013).
 26. Pan, X., Baldauf, H.-M., Keppler, O. T. & Fackler, O. T. Restrictions to HIV-1 replication in resting CD4 + T lymphocytes. *Cell Res.* **23**, 876–885 (2013).
 27. White, T. E. *et al.* The retroviral restriction ability of SAMHD1, but not its deoxynucleotide triphosphohydrolase activity, is regulated by phosphorylation. *Cell Host Microbe* **13**, 441–451 (2013).
 28. Schaller, T., Pollpeter, D., Apolonia, L., Goujon, C. & Malim, M. H. Nuclear import of SAMHD1 is mediated by a classical karyopherin $\alpha/\beta 1$ dependent pathway and confers sensitivity to VpxMAC induced ubiquitination and proteasomal degradation. *Retrovirology* **11**, 29 (2014).
 29. Antonucci, J. M. *et al.* SAMHD1-mediated HIV-1 restriction in cells does not involve ribonuclease activity. *Nat. Med.* **22**, 1072–1074 (2016).
 30. Hofmann, H. *et al.* Inhibition of CUL4A Neddylation Causes a Reversible Block to SAMHD1-Mediated Restriction of HIV-1. *J. Virol.* **87**, 11741–11750 (2013).
 31. Cribier, A., Descours, B., Valadão, A., Laguette, N. & Benkirane, M. Phosphorylation of SAMHD1 by Cyclin A2/CDK1 Regulates Its Restriction Activity toward HIV-1. *Cell Rep.* **3**, 1036–1043 (2013).

Reviewers' Comments:

Reviewer #1:

Remarks to the Author:

The authors made a significant effort to address with care all issues raised. Particularly, the results on the mutants in Figure 5 are now convincing and support the conclusions. Additionally, the authors addressed adequately the question on the proportion of sumoylated protein. Therefore, these two main concerns have been satisfactorily addressed, and the results improved the manuscript.

One minor issue: PLA experiments (Figure 1F and Supplementary Figures 2C and 7C of the revised manuscript): In these figures, measurements of PLA/IF ration (Fig 1F) and Nb dots/cell (S2C and S7C) are plotted in Box plots. The data shown here are representative of measurements of multiple cells (90-150) from one single experiment. The authors use statistical testing (T test in 1F and Anova in S2C and S7C) to prove significant difference between the groups. However, in contrast to the traditional use of these tests, the authors compare measurements of individual cells from one experiment, thus technical replicates and not biological replicates (I.E. mean of repeated experiments). This results in a high number of (technical) replicates and a high significance score. In my opinion, the use of a statistical test in this setting is misleading since it suggests a significant biological difference. However, the experiments lack biological replicates, there is obviously no biological variance in the data. The figure legends (particularly in 1F) do not describe the fact that technical replicates are compared with sufficient precision. Thus, I would suggest to simply omit the statistical test and describe the difference in the groups (from one experiment) either in a Box Plot, or/and by using a dot plot depicting distribution individual measurements more intuitively or alternatively to elaborate the respective figure legends.

Reviewer #2:

Remarks to the Author:

The authors addressed all my comments and I am satisfied with their responses. In conjunction with the suggestions made by the other reviewers, the paper has greatly improved.

Reviewer #3:

Remarks to the Author:

The manuscript by Martinat and coworkers has been extensively revised by the authors, following reviewers criticisms and suggestions. I have no further comments for improvement.

We would like to warmly thank the reviewers for assessing once more our manuscript. Their valuable feedback helped us improving substantially our work. We are glad that they appreciated our efforts to address all their concerns and found our additional results convincing. In relationship to the new issues they have addressed, a point-by- point response follows.

REVIEWERS' COMMENTS

Reviewer #1 (Remarks to the Author):

The authors made a significant effort to address with care all issues raised. Particularly, the results on the mutants in Figure 5 are now convincing and support the conclusions. Additionally, the authors addressed adequately the question on the proportion of sumoylated protein. Therefore, these two main concerns have been satisfactorily addressed, and the results improved the manuscript.

One minor issue: PLA experiments (Figure 1F and Supplementary Figures 2C and 7C of the revised manuscript): In these figures, measurements of PLA/IF ration (Fig 1F) and Nb dots/cell (S2C and S7C) are plotted in Box plots. The data shown here are representative of measurements of multiple cells (90-150) from one single experiment. The authors use statistical testing (T test in 1F and Anova in S2C and S7C) to prove significant difference between the groups. However, in contrast to the traditional use of these tests, the authors compare measurements of individual cells from one experiment, thus technical replicates and not biological replicates (I.E. mean of repeated experiments). This results in a high number of (technical) replicates and a high significance score. In my opinion, the use of a statistical test in this setting is misleading since it suggests a significant biological difference. However, the experiments lack biological replicates, there is obviously no biological variance in the data. The figure legends (particularly in 1F) do not describe the fact that technical replicates are compared with sufficient precision. Thus, I would suggest to simply omit the statistical test and describe the difference in the groups (from one experiment) either in a Box Plot, or/and by using a dot plot depicting distribution individual measurements more intuitively or alternatively to elaborate the respective figure legends.

We are grateful to reviewer#1 for his/her positive feedback and for acknowledging the quality of our responses to his/her criticisms.

We agree with the reviewer that the quantitative analysis of Figure 1f relies on the comparison of individual cells, which represent technical replicate from one experiment. Thus, we modified the figure according to his/her suggestions and omitted the statistical results.

Reviewer #2 (Remarks to the Author):

The authors addressed all my comments and I am satisfied with their responses. In conjunction with

the suggestions made by the other reviewers, the paper has greatly improved.

We thank again reviewer#2 for his/her suggestions that helped us to improve our paper and we are pleased that our revised version meets his/her expectations.

Reviewer #3 (Remarks to the Author):

The manuscript by Martinat and coworkers has been extensively revised by the authors, following reviewers criticisms and suggestions. I have no further comments for improvement.

We are grateful to reviewer#3 for his/her kind revision.